ecology, physiology

mammals, gut microbiome, amino acid metabolism, compound-specific stable isotope analysis, mutualism

**Author for correspondence:**
Seth D. Newsome
e-mail: newsome@unm.edu

# Isotopic and genetic methods reveal the role of the gut microbiome in mammalian host essential amino acid metabolism

Seth D. Newsome[1], Kelli L. Feeser[1], Christina J. Bradley[2,3], Caitlin Wolf[1], Cristina Takacs-Vesbach[1] and Marilyn L. Fogel[3,4]

[1]Department of Biology, University of New Mexico, Albuquerque, NM 87131, USA
[2]Department of Biological Sciences, Salisbury University, Salisbury MD, USA
[3]College of Natural Science, University of California Merced, Merced, CA, USA
[4]Department of Earth and Planetary Sciences and EDGE Institute, University of California Riverside, Riverside CA, USA

(iD) SDN, 0000-0002-4534-1242

Intestinal microbiota perform many functions for their host, but among the most important is their role in metabolism, especially the conversion of recalcitrant biomass that the host is unable to digest into bioavailable compounds. Most studies have focused on the assistance gut microbiota provide in the metabolism of carbohydrates, however, their role in host amino acid metabolism is poorly understood. We conducted an experiment on *Mus musculus* using 16S rRNA gene sequencing and carbon isotope analysis of essential amino acids ($AA_{ESS}$) to quantify the community composition of gut microbiota and the contribution of carbohydrate carbon used by the gut microbiome to synthesize $AA_{ESS}$ that are assimilated by mice to build skeletal muscle tissue. The relative abundances of Firmicutes and Bacteroidetes inversely varied as a function of dietary macromolecular content, with Firmicutes dominating when mice were fed low-protein diets that contained the highest proportions of simple carbohydrates (sucrose). Mixing models estimated that the microbial contribution of $AA_{ESS}$ to mouse muscle varied from less than 5% (threonine, lysine, and phenylalanine) to approximately 60% (valine) across diet treatments, with the Firmicute-dominated microbiome associated with the greatest contribution. Our results show that intestinal microbes can provide a significant source of the $AA_{ESS}$ their host uses to synthesize structural tissues. The role that gut microbiota play in the amino acid metabolism of animals that consume protein-deficient diets is likely a significant but under-recognized aspect of foraging ecology and physiology.

## 1. Introduction

Herbivores and omnivores often consume low-quality diets deficient in the amount and quality of protein that is required to maintain homeostasis and reproduce [1]. This is particularly true of animals that live in seasonal temperate environments where resources are scarce during colder periods, or in arid eco-systems characterized by low annual primary production. A plant-based diet can be low-quality for three primary reasons. First, the cell walls of many plant structural tissues primarily consist of structural compounds—cellulose, hemicellulose, and lignin—that are refractory to digestion by gastrointestinal enzymes [2]. Second, plant tissues have low-protein contents as a consequence of their high concentrations of structural compounds. Third, to deter herbivory, many plants produce secondary compounds (e.g. alkaloids and terpenoids)

**Figure 1.** Schematic of the small intestine where most protein digestion and assimilation occurs in the mammalian gastrointestinal tract. The host synthesis (HS) pathway refers strictly to non-essential amino acids ($AA_{NESS}$). Direct routing from diet ($R_D$), direct routing from microbes ($R_M$), and microbial synthesis (MS) are the three primary pathways of essential amino acid ($AA_{ESS}$) incorporation. The two microbially mediated pathways ($R_M$ and MS) could be facilitated by either lumen- or mucin-associated bacteria.

that can have both direct (toxic) and indirect (reduce digestion efficiency) negative physiological effects on animal consumers [1,3–5].

To survive and reproduce on a low-quality diet, most herbivores and omnivores maintain a community of symbiotic bacteria [6] that can synthesize metabolites that may be used by the host for energy or to build structural tissues. Mammalian herbivores are particularly reliant on their gut microbiome to help digest the complex carbohydrates they consume and are often classified into two groups based on their digestive physiology–foregut and hindgut fermenters. In mammals, the highest densities of intestinal bacteria are typically found in specialized organs (e.g. rumen, cecum) where recalcitrant plant biomass is fermented by anaerobic microbes to form short-chain fatty acids more easily assimilated by the host as either an energy source or as a substrate for the *de novo* synthesis of other metabolites like non-essential amino acids. It has even been hypothesized that the coevolution between mammalian herbivores and their intestinal bacteria was a factor in the Cenozoic mammal radiation [7,8], especially during the past 10 million years when particularly low-quality $C_4$ grasses evolved and spread throughout the globe [9]. In addition to the degradation of plant structural carbohydrates, more recent research has shown that intestinal bacteria also facilitate the digestion of plant secondary compounds, which has likely influenced the adaptation and evolution of mammalian lineages that live in nutrient-poor environments where the consumption of plants containing high concentrations of secondary compounds may be crucial resources for survival [4,5].

The two most dominant bacteria phyla found in the intestines of mammalian herbivores are Firmicutes and Bacteroidetes [6], and their co-dominance can be explained by their occupation of different functional niches [10]. Bacteroidetes are more closely associated with the degradation of complex glycans [11,12], while Firmicutes rely more on the fermentation of simpler polysaccharides and short-chain fatty acids [11,13,14]. Thus, community dynamics are in large part determined by host diet composition [11,15]. Eukaryotic microbes (e.g. fungi) are also found in mammalian guts, though their functional roles are unclear [16]. To date, most studies have focused on the role of gut microbiota in host carbohydrate metabolism, which has important implications for human health [17].

Exploring the complex functional role(s) that the gut microbiome plays in the protein metabolism of mammals is a burgeoning field. Eukaryotic hosts can acquire non-essential amino acids in several ways (figure 1): (i) route them directly or indirectly (via the gut microbiome) from dietary protein, (ii) synthesize them *de novo* with carbon from dietary carbohydrates, lipids, or dietary amino acids (AAs) catabolized during metabolism [18–20], and/or (iii) route them directly from non-essential AAs produced by the gut microbiome from dietary carbohydrate and/or lipid precursors [19,21,22]. By contrast, the $AA_{ESS}$ needs of eukaryotes must be routed directly from dietary protein or supplied by the gut microbiome.

The potential sources of AAs and the pathways by which they can be assimilated by the host are numerous (figure 1). Dietary protein is first hydrolysed during digestion to produce short peptides and AAs, which can be directly assimilated by the host or gut bacteria and catabolised for energy, or be used to synthesize proteins to maintain/grow tissues and (microbial) cells. First, the host can synthesize non-essential AAs using carbon derived from dietary carbohydrates or lipids [18–20,23], which is depicted by the host synthesis (HS) pathway in figure 1. Second, dietary $AA_{ESS}$ and non-essential AAs can be routed by the host ($R_D$ pathway) or associated gut bacteria ($R_M$ pathway). Third, bacteria can also synthesize *de novo* all AAs required for protein biosynthesis [24], which is depicted by the MS (microbial synthesis) pathway in figure 1. The carbon needed for *de novo* synthesis of $AA_{ESS}$ by gut bacteria may be sourced from non-protein dietary macromolecules (carbohydrates or lipids) or catabolized dietary protein and thus is potentially important in situations where dietary protein is limited; even some non-essential AAs can be conditionally essential in rapidly growing animals [20]. Gut microbes may catabolize host intestinal tissues in the form of glycans (saccharides) or glycoproteins present in the mucin [25,26] and use these substrates to either synthesize AAs *de novo* or directly route them into their cells.

While the role gut microbiota play in the protein metabolism of wild animals has not been systematically explored, lab experiments utilizing isotopically labelled substrates (e.g. $^{15}NH_4Cl$) show that the gut microbiome can provision its host to various degrees with $AA_{ESS}$ [27–31]. For mammals,

studies on a limited set of AAs show that gut microbes can contribute from less than 5% to as much as 80% of the $AA_{ESS}$ in the host plasma pool [28]. Utilizing a different method that analysed the carbon isotopes ($\delta^{13}C$) of individual AA at natural abundances, our experiment on an omnivorous fish found that the carbon in the $AA_{ESS}$ in the muscle of tilapia (Oreochromis niloticus) fed diets low in protein (less than 4%) was sourced from dietary carbohydrates [19]; calculations indicate that intestinal bacteria account for approximately 50% of some of the $AA_{ESS}$ in tilapia muscle when fed low-protein diets. A similar approach has been applied to phytophagous insects that persistently consume AA-deficient diets [21,22,32]. Overall, data from both labelled and natural abundance isotope-based experiments suggest that intestinal bacteria can play a significant role in the protein metabolism of their host.

To understand the functional significance of the microorganisms responsible for this important and likely common biochemical pathway in wild omnivorous and herbivorous animals whose growth, overall health, homeostasis, and reproduction are often limited by protein availability, we conducted a controlled feeding experiment with Mus musculus that were fed one of four diets that varied in the content of a $C_3$-based ($\delta^{13}C$: −26.5‰) protein and $C_4$-based ($\delta^{13}C$: −12.0‰) carbohydrates. We then used 16S rRNA gene analysis of mice ceca, AA $\delta^{13}C$ analysis, and a series of models to estimate how the microbial contribution of each of six major $AA_{ESS}$ to muscle protein varied across diet treatments. We predicted that the composition of the gut bacterial community would be sensitive to dietary protein: carbohydrate content. We also predicted that microbial contribution of individual $AA_{ESS}$ would increase with decreasing dietary protein content. Our novel combination of next-generation genetic sequencing and isotope analysis of a large suite of individual $AA_{ESS}$ enabled us to quantify the degree to which the gut microbiome contributes on a compound-specific level to the $AA_{ESS}$ pool used by its host to synthesize skeletal muscle.

# 2. Material and methods

## (a) Experimental design

Sixty recently weaned approximately 3- to 4-week-old Swiss Webster house mice (Mus musculus) purchased from Charles River Laboratories (Wilmington, MA, USA) and randomly divided into four treatment groups ($n = 10$/group) were housed in four plastic cages at 25°C with a 12 L : 12D photoperiod. Each treatment received one of four diets that varied inversely in $C_3$-based protein ($\delta^{13}C = −26.5‰$) versus $C_4$-based carbohydrate content, which included sucrose ($\delta^{13}C = −12.2‰$) that remained in constant proportion (36%) among treatments, and cornmeal ($\delta^{13}C = −12.0‰$) that varied inversely with protein content (electronic supplementary material, table S1). Note that the cornmeal used in our experiment was 10% protein by weight. All work was conducted with the approval of the University of New Mexico Institutional Animal Care and Use Committee (#A-4023-01). While lab-reared mice typically host the same suite of microbial phyla in their guts as wild rodents, they support higher relative proportions of Firmicutes to Bacteroidetes [33] and as such are not an ideal analogue for their wild counterparts.

## (b) Gut microbiome community composition

Mouse ceca were dissected under sterile conditions from five randomly selected mice from each diet, preserved in equal volumes of sucrose lysis buffer [34] and stored at −20°C until DNA extraction was performed. Samples were homogenized by vortexing, and DNA was extracted from 500 µl subsamples using a variation of the cetyltrimethylammonium bromide (CTAB) method [35]. Dual-index paired-end amplicon sequencing of 16S rRNA genes was performed as previously described [36,37] using V1-3 universal bacterial primers 28F (5′-GAG TTT GAT CNT GGC TCA G-3′) and 519R (5′-GTN TTA CNG CGG CKG CTG-3′) on an Illumina MiSeq. Quantitative Insights into Microbial Ecology [38], Dada2 [39], and the Vegan library in R [40] were used to analyse the 16S rRNA gene sequences. The sequence data from this study are available through the National Center for Biotechnology Information (NCBI) Sequence Read Archive (accession numbers SRX6959962 through SRX6959981).

## (c) Amino acid $\delta^{13}C$ analysis

Mouse muscle tissue was lipid-extracted, hydrolysed, derivatized, and corrected for carbon added during derivatization [19]; additional details are reported in the electronic supplementary material. Derivatized samples were injected into a Thermo Scientific Trace GC Ultra gas chromatograph for separation with a BPX5 60 m, 0.32 mm ID, 1.0 µm film thickness column (SGE Analytical Science), then converted to $CO_2$ via a Thermo Scientific IsoLink combustion interface and analysed with a Thermo Scientific Delta V Plus isotope ratio mass spectrometer at the University of California Merced (Merced, CA).

## (d) Mixing models

We applied a linear mixing model that incorporated weight per cent diet proportions and associated $AA_{ESS}$ concentrations to estimate the availability of dietary $AA_{ESS}$ sourced from both casein and cornmeal, which was 10% protein by weight. This first model (Model #1) generated treatment-specific $\delta^{13}C$ values for $AA_{ESS}$ that could be directly routed from diet, which for most diet treatments was overwhelmingly casein but did include a significant cornmeal component in the low-protein diet treatment. Second, we used another linear mixing model to estimate the proportion of each $AA_{ESS}$ in mouse muscle that was directly routed from dietary protein versus that sourced from dietary carbohydrates (cornmeal and sucrose). This second model (Model #2) used the $\delta^{13}C$ values generated by the first as one potential source and predicted $\delta^{13}C$ values of $AA_{ESS}$ synthesized by microbes from dietary carbohydrates that were estimated with AA-specific isotopic fractionation factors reported in [19,41] as the second source. We then used $AA_{ESS}$ $\delta^{13}C$ values of mouse muscle and our two end member sources in a linear mixing model to estimate the proportion of $AA_{ESS}$ in mouse muscle that was ultimately derived from microbial synthesis versus directly routed from dietary protein.

## (e) Contribution of the microbial community composition to host tissue synthesis

The degree to which microbial community composition impacted host muscle $AA_{ESS}$ $\delta^{13}C$ was investigated by comparing operational taxonomic unit (OTU)-level Bray–Curtis dissimilarities among samples to $AA_{ESS}$ $\delta^{13}C$ values using canonical redundancy analysis (RDA). The significance of results was assessed through 1000 permutations.

## (f) Relative $AA_{ESS}$ supply and demand

We used a combination of data on dietary protein content and published estimates of digestibility [42] and mucosal catabolism [43] to calculate the overall supply of individual $AA_{ESS}$ and Asp for host metabolism. We also used published estimates of AA requirements for rapidly growing mice [44,45] to calculate metabolic demands. To

estimate AA supply that accounted for both AA-specific estimates of digestibility as well as catabolism in the intestinal mucous presumably by the gut microbiome, we first converted the daily ration into an experiment ration. Finally, we accounted for the proportion of dietary AAs catabolized by mucosal microbiota in non-ruminant mammals fed diets in which casein was the only protein [43,46]; we used estimates of mucosal catabolism reported in [43] of 30% for Ile, 40% for Leu and Val, 45% for Phe, 50% for Lys, and 60% for Thr. Additional details on supply and demand calculations are reported in the electronic supplementary material. A variety of factors such as diet and/or gut microbiome community composition can influence our AA supply calculations, thus we attempted to use published information for non-ruminant mammals that mimicked our experimental diets in terms of protein type and content.

## 3. Results

### (a) Mice growth

The rate of mass gain did not vary among diet treatments between Week 4 and Week 15 (electronic supplementary material, figure S1). We assume that muscle tissue harvested at the end of our feeding experiments (Day 120) was in steady state with diet because of the observed increases in body mass, and the results of previous feeding experiments that estimated carbon isotope incorporation rates for muscle of adult mice [47], which have slower rates than rapidly growing juveniles.

### (b) Gut microbiome community composition

Gut bacterial diversity increased with dietary protein content (electronic supplementary material, figure S3) and community composition varied among the individual ceca samples by diet treatment (figure 2a); samples clustered separately by diet (ANOSIM R statistic = 0.457, $p = 0.001$) when analysed using Bray–Curtis distances (figure 2a). Random Forests, a machine learning algorithm used for classification, could distinguish samples from the diet treatments with 95% accuracy. The ratio of baseline error to observed error was 15; a minimum ratio of 2 is expected for factors that can be accurately predicted [48].

At the phylum level, the mouse gut microbiota was dominated by at least 90% Firmicutes and Bacteroidetes. Overall, Firmicutes dominated the OTU-level diversity and represented six times more OTUs than the Bacteroidetes. The relative proportion of these two phyla correlated ($\rho = 0.53$, $p = 0.017$) to diet treatment, with the most Firmicutes found in mice fed low-protein/high-carbohydrate diets (figure 2b). Specifically, Firmicutes were most abundant (82.5% and 66.2%) in Diet #3 (12P:55C) and Diet #4 (9P:75C), respectively. By contrast, the abundance of Firmicutes relative to Bacteroidetes was more even in Diets #1 (40P:40C) and #2 (21P:45C) with 49.5 and 45.5% Firmicutes and 46.9 and 47.6% Bacteroidetes, respectively (figure 2c). At the family level (figure 2d), microbial communities were primarily composed of Lactobacillaceae, Lachnospiraceae, Ruminococcaceae (Firmicutes), Bacteroidaceae, Porphyromonadaceae, Rikenellaceae, S24-7 (Bacteroidetes), Alcaligenaceae (Proteobacteria), Deferribateraceae (Deferribacteres), and Verrucomicrobiaceae (Verrucomicrobia). OTUs from these families contributed most significantly to the classification of samples by Random Forests among the high- and low-protein diets (figure 3). A majority (79%) of the OTUs and 71% of amplicon sequence variant (ASVs) in this experiment were not classified below family despite using multiple databases for taxonomic classification.

### (c) Amino acid $\delta^{13}C$

Figure 4 shows $AA_{ESS}$ $\delta^{13}C$ values of mice muscle, as well as end member values for $AA_{ESS}$ from dietary protein and microbial $AA_{ESS}$ synthesized *de novo* with carbon from dietary carbohydrates for the four diet treatments; see electronic supplementary material, table S4 for mean (±SD) $\delta^{13}C$ values. In general, mice muscle $AA_{ESS}$ $\delta^{13}C$ values were more similar to those in dietary protein for the high-protein diets (figure 4a) than low-protein diets (figure 4b). Most mice muscle $AA_{ESS}$ in the high-protein diet treatments (figure 4a) had $\delta^{13}C$ values that were closer to those of dietary protein (open blue circles), while mice muscle $AA_{ESS}$ in the low-protein treatments (figure 4b) was more intermediate between dietary protein (open red/orange circles) and $AA_{ESS}$ synthesized *de novo* by gut microbiota from dietary carbohydrates (open grey circles).

Mixing models confirmed that $AA_{ESS}$ sourced from gut microbiota make an important contribution to those used by mice to synthesize muscle tissue (figure 5). As expected, microbially derived $AA_{ESS}$ made a larger contribution to mice fed low-protein versus high-protein diets, especially in the low-protein diet where contributions of Val and Ile to mice muscle were 60% and 40%, respectively. Estimated microbial contributions of $AA_{ESS}$ to mice muscle were between 10 and 40% for nearly all diet treatments, with the exception of Thr and Lys in the diets containing 21 and 40% protein, and Phe in the low-protein diet. There was no significant effect between eigenvalues derived from microbial OTU tables and $AA_{ESS}$ $\delta^{13}C$ values within Diets #1 (40P:40C) and #2 (21P:45C) ($p > 0.05$), but the Diet #3 (12P:55C) model had an $R_a^2$ value of 0.37 (Akaike Information Criteria (AIC) = 12.2, $F$-value = 3.37, $p = 0.008$) and the Diet #4 (9P:75C) model had an $R_a^2$ value of 0.43 (AIC = 13.5, $F$-value = 3.99, $p = 0.008$).

### (d) $AA_{ESS}$ supply and demand

Our estimates show that dietary supply outweighs or closely matched maximum metabolic demand for four of six $AA_{ESS}$ in diet treatments containing ≥12% protein (figure 6). Notable exceptions were Thr and Phe, where demand was higher than supply by 20–200%. In diet treatments containing 21% and 40% protein content, supply was consistently 50–200% higher than estimated demand for all $AA_{ESS}$ except Thr, where only the high-protein diet provided more supply than estimated demand. Lastly, the dietary supply of aspartic acid (Asp), an important intermediary for the *de novo* synthesis of Thr and Lys, was much lower than metabolic demand for all diet treatments after accounting for catabolism by intestinal mucosa (figure 6).

## 4. Discussion

Homeostasis in wild animals is often limited by dietary protein content since animals are largely constructed from AAs (approx. 60%, [51]) and thus have high-protein demands that often outweigh intake. By focusing on $AA_{ESS}$, our study could quantify the microbial contribution of these compounds that are needed by the host to build and maintain structural tissues like muscle. We observed diet-associated changes in the microbial contribution of $AA_{ESS}$ used by mice to synthesize muscle, and a significant relationship between gut microbial composition and the $\delta^{13}C$ values of $AA_{ESS}$ in the muscle of

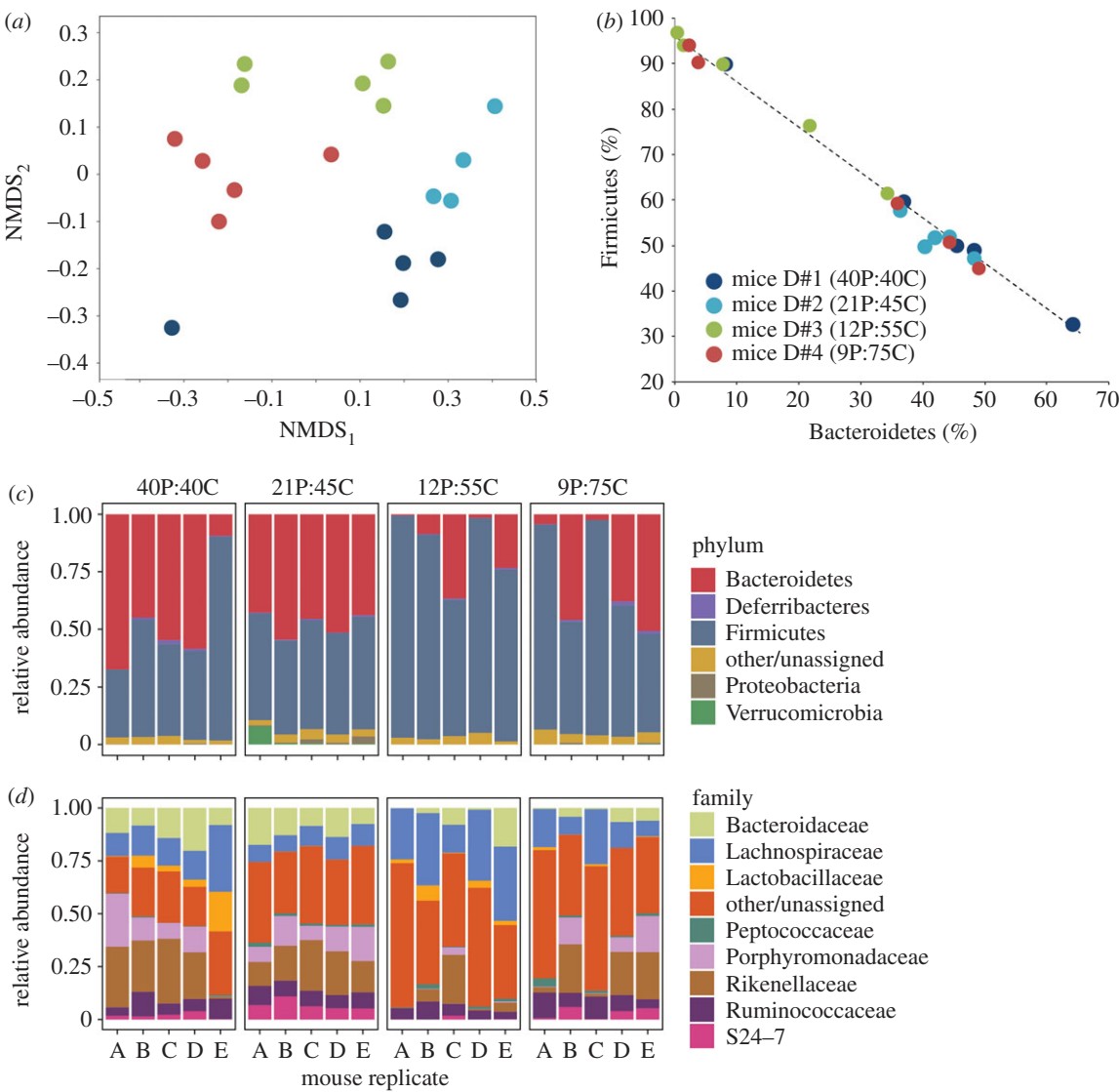

**Figure 2.** (a) Bray–Curtis non-metric multidimensional scaling (NMDS) plot of the microbial composition of ceca in mice fed one of four diets ($n = 5$ mice per treatment) that varied in per cent protein (P) and carbohydrate (C) content; see legend in Panel b for dietary proportions. (b) Relative abundance (%) of Firmicutes and Bacteroidetes in mice ceca fed these same four diets. Taxa relative abundance at the phylum (c) and family (d) resolution by diet treatment. Data have been filtered to taxa with at least 1% average relative abundance. (Online version in colour.)

mice fed low-protein diets. Predictably, $\delta^{13}$C values of $AA_{ESS}$ in muscle increased with increasing $^{13}$C-enriched dietary carbohydrate and decreasing $^{13}$C-depleted dietary protein content (figure 4), which clearly shows that the gut microbiome is supplying its host with $AA_{ESS}$ synthesized *de novo* from dietary carbohydrates. Intuitively, mixing models show that microbially derived $AA_{ESS}$ made a larger contribution to the muscle synthesized by mice fed low- versus high-protein diets. Unexpectedly, $\delta^{13}$C data for several $AA_{ESS}$ suggest a significant microbial contribution to the pool of $AA_{ESS}$ used by mice to synthesize muscle even when mice were fed diets containing 40% protein.

## (a) Influence of diet on gut microbiome

The vertebrate gut microbiome is dominated by two main bacterial phyla, the Firmicutes and Bacteroidetes [6,52] and an inverse relationship has been previously observed in the relative abundance of these two phyla in response to diet. In humans, a high proportion of Bacteroidetes is associated with a plant-based diet, whereas Firmicutes dominate when the host consumes more simple carbohydrates and fats [53].

Similar trends have been found among diverse animals [50]. Although Firmicutes are largely beneficial microbes, their relative abundance compared to Bacteroidetes may be taken as an indicator of host health including metabolic disorders and cognitive flexibility [49,50,54]; although the Bacteroidetes have also been implicated in disease [55]. Previous work on the mammalian gut microbiome has focused on the role of carbohydrate in modulating microbiome community composition and host health (e.g. [7,46,53,56]). The role of dietary protein in gut microbiome composition is less understood, although nitrogen plays an important role in host–microbiome metabolic interactions [57]. Our experimental results agree with previous reports as both gut microbiome community composition (figure 2a) and the ratio of Bacteroidetes versus Firmicutes (figure 2b) were strongly influenced by diet. Firmicutes were a greater proportion (approx. 60–98%) of the communities in the mice fed the low-protein (high-carbohydrate) diet, whereas the relative abundance between these two phyla was more even in the high-protein diets.

Proteobacteria, Deferribacteres, and Verrucomicrobia were also detected (figure 2c) and played a significant role in classifying the gut communities according to diet treatment (figure 3).

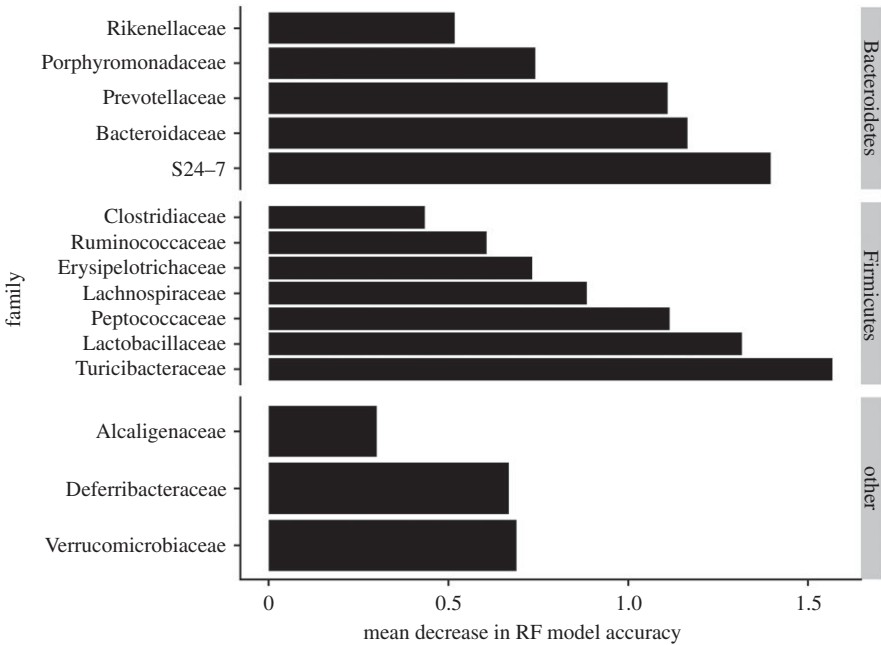

**Figure 3.** Families of OTUs that contributed most significantly to the classification of samples among the high- and low-protein diets in the Random Forests analysis shown as mean decrease in model accuracy.

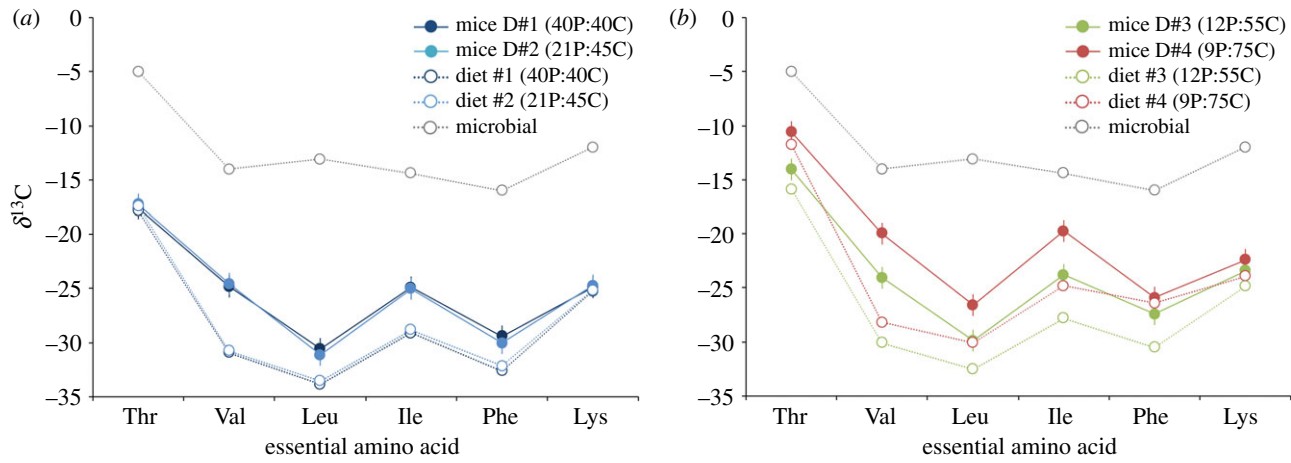

**Figure 4.** $AA_{ESS}$ $\delta^{13}C$ values in muscle (solid circles) and dietary protein (open circles) for mice fed four diets that varied in protein versus carbohydrate content; error bars for mice muscle represent standard error and sample size is six mice per treatment. Numbers in parentheses in the legend are per cent protein (P) versus carbohydrate (C) contents of each diet treatment, while red open circles represent estimated $\delta^{13}C$ values of amino acids synthesized *de novo* by gut microbes using dietary carbohydrates. Lines connecting data from each experiment are for graphical clarity and do not denote statistical relationships. (Online version in colour.)

In humans, gut Proteobacteria populations are often associated with disease, but were rare in our experiment and were most abundant in the high-protein diets. Members of the Deferribacteres and Verrucomicrobia are generally not abundant in animal guts, however, the genera *Mucispirillum* and *Akkermansia* from the these phyla, respectively, were among the few OTUs identified to species in this experiment. These species have been associated with the mucin of animals [25,58], although their roles in host health are still being investigated [59]. The degree to which diet influences community changes may be driven by the relative abundance of dietary nutrients versus host gastrointestinal secretions is not known [25,26]. Thus, the modulation of endogenous versus exogenous nutrients creates opportunities for competitive interactions over niche resources [57] which deserves further investigation.

## (b) $AA_{ESS}$ supply and demand
Another important factor influencing the microbial contribution of $AA_{ESS}$ to host protein metabolism is the balance between the availability of $AA_{ESS}$ in dietary protein and the host's metabolic demand of $AA_{ESS}$ required for tissue growth and maintenance. Combining the two sources of dietary protein (casein and cornmeal) and comparing their relative supply across diet treatments directly to their demand based on data for rapidly growing mice [44,45] showed that dietary supply was greater than demand for five of the six $AA_{ESS}$ we measured in all but the low-protein diet (figure 6). Thr was the only $AA_{ESS}$ for which demand significantly outweighed supply by more than twofold in diets containing ≤12% protein. Likewise, Lys supply in the low-protein diet treatment was only 60% of demand, but closely matched demand in the diet containing 12% protein. Even though the microbial contribution of Thr and Lys was relatively low in comparison to most other $AA_{ESS}$, we observed approximately 3–6-fold increases in the microbial contribution of these two $AA_{ESS}$ as dietary protein content decreased from 21 to 12% (figure 5). Likewise, we also observed significant increases in the microbial contribution

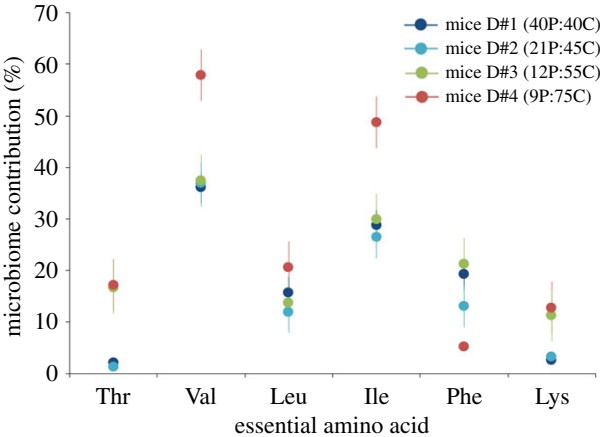

**Figure 5.** Estimated proportion of AA$_{ESS}$ in mouse muscle that was synthesized *de novo* by gut microbiota using carbohydrate dietary sources (sucrose and cornmeal). Numbers in parentheses in the legend are per cent protein (P) versus carbohydrate (C) contents of each diet treatment. Error bars represent standard error based on estimated microbial contributions for six individual mice per treatment. (Online version in colour.)

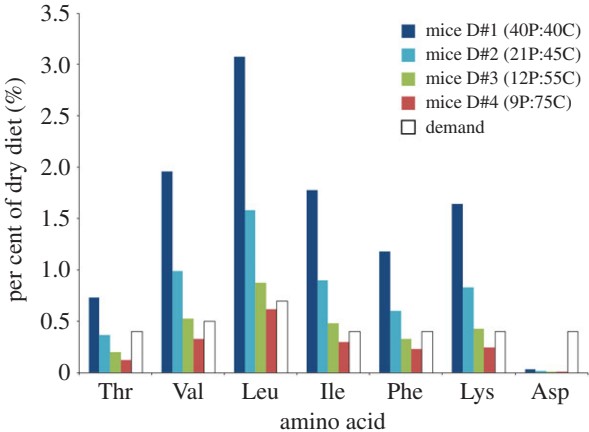

**Figure 6.** Estimates of the relative dietary supply versus demand of AA$_{ESS}$ and Asp as a percentage in dry food; data for demand are from [49,50]. Numbers in parentheses in the legend are per cent protein (P) versus carbohydrate (C) contents of each diet treatment. (Online version in colour.)

of Val and Ile to muscle synthesis, however, only when mice were fed the low-protein diet in which supply of these two AA$_{ESS}$ was only 65–75% of demand (figure 6). These patterns confirm that relative dietary supply and metabolic demand of AA$_{ESS}$ are important factors influencing the microbial contribution of AA$_{ESS}$ to host protein metabolism.

One of the more unexpected results was that estimates of the microbial contribution of some AA$_{ESS}$ to mouse muscle synthesis were surprisingly high and did not vary with dietary protein supply. In four of the six AA$_{ESS}$ measured (Val, Ile, Leu, Phe), isotope-based estimates of the gut microbial contribution to mouse muscle synthesis varied from approximately 15% to nearly 40% when mice were fed high-protein diets that far exceeded their requirements for optimal growth (figure 6; [44,45]). We observed a similar pattern in both direction and magnitude in a previous experiment [23], where mouse muscle Ile and Val had $\delta^{13}C$ values 4–7‰ higher than the corresponding AAs in their diet containing adequate protein content that more than met demands (figure 4).

Another important consideration related to supply is availability of other compounds and metabolic intermediaries that

are precursors to AA$_{ESS}$ synthesis by microbes. For example, aspartate (Asp) is required to synthesize aspartate semialdehyde, an intermediary in both Thr and Lys synthesis. Dietary supply of Asp was much lower than metabolic demands in all diet treatments [(figure 6); 44], thus *de novo* synthesis of Thr and Lys by the gut microbiome may be limited by the availability of this non-essential AA.

Lastly, direct comparisons between gut microbiome community composition and muscle AA$_{ESS}$ $\delta^{13}C$ values, our proxy for microbial contribution of AA$_{ESS}$ to host tissue synthesis, showed no significant relationship in the two diets that contained high amounts of protein. By contrast, the relationship between gut microbial composition and host muscle AA$_{ESS}$ $\delta^{13}C$ values was significant and the explainable variation increased from 37 to 42% with decreasing dietary protein content from 12 to 9%. These results support our hypothesis that the microbial contribution to host muscle synthesis increases when the supply of exogenous protein is low.

## (c) Metabolic costs

Another factor that could influence the microbial contribution of AA$_{ESS}$ is related to metabolic costs associated with their *de novo* synthesis by the gut microbiome. There are costs in terms of energy (ATP and NADPH) as well as the number of steps and types of intermediaries involved in specific biochemical pathways needed to synthesize AA$_{ESS}$ from carbohydrates [60]. Variation in the estimated microbial contribution among AA$_{ESS}$ (figure 5) suggests that some forms may be less costly for the gut microbiome to synthesize than others. For example, the microbial contribution of Val and Ile was greater than 30% regardless of dietary protein content, while dietary supply of these AA$_{ESS}$ were 2–4-fold higher than demand in the two diets containing greater than 20% protein (figure 6). By contrast, the estimated microbial contributions of Leu and Phe were relatively low and invariant among diet treatments. For Leu, dietary supply exceeded relative demand in all diets except the low-protein treatment, but supply of Phe was lower than demand in the two diets containing ≤12% protein (figure 6). When combined with estimates of supply and demand, these patterns suggest that metabolic costs associated with *de novo* synthesis of AA$_{ESS}$ influenced the observed variation in microbial contribution among individual AA$_{ESS}$.

Biochemical pathways for *de novo* synthesis of AA$_{ESS}$ are often grouped into precursor families, each of which shares a similar primary precursor, intermediaries, and/or enzymes. The six AA$_{ESS}$ analysed here are synthesized via a number of steps from two primary precursors associated with glycolysis (pyruvate and phosphoenolpyruvate) or oxaloacetate, a precursor that is an intermediary in the tricarboxylic acid (TCA) cycle. Three of the four AA$_{ESS}$ (Val, Ile, and Leu) that showed high (15–40%) gut microbiome contributions in mice fed high-protein diets are branch-chained AAs that play vital roles in protein synthesis and catabolism, cell signalling, and the metabolism of glucose [61]. Val, Ile, and Leu are synthesized via 4–7 steps from pyruvate, but generally require less energy to build in comparison to AA$_{ESS}$ in the other two precursor families [58]. Val requires only 5 steps and 2 NADPH from pyruvate and thus it is not surprising that microbial contributions were highest for this AA$_{ESS}$, even when mice were fed high-protein diets. By contrast, the high microbial contributions of Ile are difficult to explain

given how metabolically expensive it is to make (2 ATP, 5 NADPH, 11 steps from pyruvate) relative to Val, and that relative dietary supply of Ile exceeded metabolic demands in all treatments except the diet with the lowest protein content (figure 6). Estimated microbial contributions of Leu were generally lower than for other AA$_{ESS}$ in the pyruvate precursor family, even though Leu requires only 2 NADPH but a total of 8 steps from pyruvate to synthesize. The relatively high supply of Leu available directly from diet in most treatments (figure 6) may have decreased the relative contribution of this AA$_{ESS}$ from the gut microbiome.

Given its complex ring structure and large number of steps ($n = 11$) needed to synthesize it from phosphenolpyruvate, Phe is costly to make in comparison to most of the other AA$_{ESS}$ we measured. Thus, it's not surprising that microbial contributions were relatively low in comparison to AA$_{ESS}$ that belong to the pyruvate precursor family. Moreover, in contrast with other AA$_{ESS}$, the observed isotopic pattern (figure 4) and estimated microbial contribution (figure 5) of Phe in mouse muscle did not vary inversely with dietary supply (figure 6). This finding suggests that Phe was routed to a greater degree from dietary protein when mice were fed inadequate amounts of this AA$_{ESS}$ to meet demands, perhaps because Phe is so costly to synthesize.

The estimated microbial contributions for Thr and Lys, which are derived from the TCA cycle intermediary oxaloacetate, were generally much lower than AA$_{ESS}$ synthesized from pyruvate. We suggest this pattern is driven in part by the high costs of synthesizing these two AA$_{ESS}$. Thr synthesis occurs in only five steps from oxaloacetate but is relatively energy intensive and requires 2 ATP and 3 NADPH. Lys synthesis is even more complicated and can occur via four unique pathways requiring 8 to 10 steps from oxaloacetate, and 3 ATP and 4 NADPH to make a single molecule.

Although our estimates on the exact percentage of microbially sourced AAs contain several assumptions, we can categorically state that if the $\delta^{13}$C of an AA$_{ESS}$ in mouse muscle is not equal to that in its diet, it must have been synthesized by microbial processes. Data published from several microbial experiments [41,62,63] show that isotope fractionations by a single type of microbe are as complex as those in eukaryotes. Our experiment identified thousands of bacterial OTUs in the gut microbiome of mice, each one of which could be metabolizing dietary macromolecules slightly differently. At a broad scale, the Firmicutes that were in greater abundance in the guts of mice fed a low-protein diet could very well produce a different microbial isotopic signal than the Bacteroides, which were in greater abundance in mice fed high-protein diets. By virtue of the fact that Bacterioides were the preferred microbes in the guts of mice fed high-protein diets, this finding may imply that they were very efficient in turning both dietary protein (casein) and sugars into AAs, thereby resulting in higher proportions of microbially derived AA$_{ESS}$ (20–60%, figure 5) than we might have originally thought was reasonable. The opposite could be true with the Firmicutes, which target simple carbohydrates

presumably for energy [6] but might be less efficient at synthesizing AA$_{ESS}$ at levels that could impact the AA$_{ESS}$ budget of their host. We anticipate that a greater understanding of the complex interactions among diet, gut microbial species, and host supply and demand will require further study of wild species and a combination of genetic and isotopic approaches to uncover linkages between microbiome composition and function [63].

## 5. Conclusion

Overall, our data show that the degree of AA$_{ESS}$ synthesized by the gut microbiome from dietary carbohydrates and used by the host to build structural tissues (muscle) is strongly correlated with the composition of the microbiome community. Specifically, mice used a significant proportion of microbially derived AA$_{ESS}$ and had bacterial microbiomes dominated by Firmicutes when fed diets low in protein but full of simple carbohydrates (sucrose). Whether Firmicutes are responsible for synthesizing these AA$_{ESS}$ is currently unknown, however, the high proportions of microbially derived AA$_{ESS}$ observed in mouse muscle shows how important the gut microbiome as a whole is for mediating AA metabolism in the host. Because microbes can affect the isotopic composition of AAs in animal tissues, ecologists who increasingly use the carbon isotope fingerprints of AAs to determine the ultimate sources of primary production to an animal's diet [41] need to be aware that such microbial influence needs to be considered. In addition, nitrogen isotope values of AAs are now considered reliable tracers of trophic level, but such estimates are based on the premise that some AAs are passed up the food chain without significant isotopic modification [64]. Clearly, we need to rethink these ideas based on the fact that all animals have a gut microflora, which likely contribute to the AA$_{ESS}$ budget used to synthesize their tissues. As we quantify the role of the gut microbiome in the protein metabolism of more host organisms, the phrase 'you are what you eat' becomes more appropriate than previously considered, since our study shows that a combination of both exogenous as well as endogenous sources of nutrition derived from the gut microbiome may influence host fitness.

Data accessibility. The DNA sequence data reported in this paper are available from the Sequence Read Archive at NCBI under BioProject accession number PRJNA576270. All isotope data reported in this paper are available in the electronic supplementary material.

Competing interests. We declare we have no competing interests

Acknowledgements. The DNA sequencing for this project was completed with the assistance of Darrell Dinwiddie at the UNM Health Science Center. David Araiza (UC Merced) assisted with $\delta^{13}$C measurements. This work was supported by the University of New Mexico Research Allocation Committee, College of Natural Sciences at the University of California Merced, and National Science Foundation awards to SD Newsome and C Takacs-Vesbach (IOS-1755402), and ML Fogel (IOS-1755353).

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
