## [Reviewer comments · Proceedings of the Royal Society B: Biological Sciences]

Review History

RSPB-2019-1551.R0 (Original submission)

Review form: Reviewer 1 (Paul Koch)

Recommendation

Major revision is needed (please make suggestions in comments)

Scientific importance: Is the manuscript an original and important contribution to its field?

Excellent

General interest: Is the paper of sufficient general interest?

Excellent

Quality of the paper: Is the overall quality of the paper suitable?

Excellent

Is the length of the paper justified?

Yes

Should the paper be seen by a specialist statistical reviewer?

No

Do you have any concerns about statistical analyses in this paper? If so, please specify them explicitly in your report.

Yes

It is a condition of publication that authors make their supporting data, code and materials available - either as supplementary material or hosted in an external repository. Please rate, if applicable, the supporting data on the following criteria.

Is it accessible?

Yes

Is it clear?

Yes

Is it adequate?

Yes

Do you have any ethical concerns with this paper?

No

Comments to the Author

There are many, relatively minor comments on the manuscript. The two large comments are the following.

- 1) The equation reported for Model 1 is incorrect. It is a linear mixing model, not a concentration dependent model. Report the appropriate equation and use it to calculate the dietary end member.
- 2) Explain the deep assumptions underpinning your approach to modeling the microbial end member. In particular, why isn't dietary cellulose (C3) being considered a potential C substrate for microbes? If it is a substrate, then the C end member used here is wrong.
- 3) I found the entire discussion of catabolism from the gut mucosa pretty much intelligible.

Review form: Reviewer 2

Recommendation

Reject - article is scientifically unsound

Scientific importance: Is the manuscript an original and important contribution to its field?

Acceptable

General interest: Is the paper of sufficient general interest?

Acceptable

Quality of the paper: Is the overall quality of the paper suitable?

Marginal

Is the length of the paper justified?

Yes

Should the paper be seen by a specialist statistical reviewer?

No

Do you have any concerns about statistical analyses in this paper? If so, please specify them explicitly in your report.

No

It is a condition of publication that authors make their supporting data, code and materials available - either as supplementary material or hosted in an external repository. Please rate, if applicable, the supporting data on the following criteria.

Is it accessible?

Yes

Is it clear?

Yes

Is it adequate?

Yes

Do you have any ethical concerns with this paper?

No

Comments to the Author

This paper set out to quantify the contribution of gut microbes to the essential amino acid metabolism. This physiological function has been greatly overlooked in the microbiome literature, and so the techniques used here are promising. However, there were several large issues with this experiment.

Larger Issues

Relevance – the introduction of this paper heavily discusses herbivorous animals, but the experiment was conducted in laboratory mice. These animals are highly inbred, and also vendor-purchased mice have extremely different microbial communities from their wild counterparts, which also influences aspects of their physiology.

[https://www.cell.com/cell/pdf/S0092-8674\(17\)31065-6.pdf](https://www.cell.com/cell/pdf/S0092-8674(17)31065-6.pdf)

Related, the paper doesn't discuss much of what has been demonstrated in regards to microbial provisioning of essential amino acids. These processes have been studied before using labeled substrates, and could be mentioned a bit more in the introduction. Better set up how this paper is novel from what has been done (right now the intro just says that this process is poorly understood). I think the info on fiber degradation could be extensively reduced to allow for more discussion of amino acid- related processes.

Communal housing of mice – it is not widely known that co-housed mice share aspects of their microbiome with each other. This cohousing likely reduced inter-individual variation across your treatment groups, and also results in the fact that these individuals are not completely independent units. <https://academic.oup.com/femsre/article/40/1/117/2467665>

Individual physiology not measured – Much of the details used in the (food intake, digestibility, etc) seem to be taken from the literature. However, I wonder how these variables may have differed across treatment groups, and may have influenced the results and conclusions. For example, the digestive system can change its function to promote optimal digestion/absorption: <https://royalsocietypublishing.org/doi/full/10.1098/rspb.2009.2045> I'd like if the authors could

explicitly state what variables in their analysis are being taken from the literature, and how variation here could alter their results.

Microbiome data poorly analyzed and poorly connected to amino acid metabolism – The only microbiome results presented are differences in Firmicutes+Bacteroidetes. With microbiome sequencing there is lots of deeper information that can be gleaned. Also, many studies are now connecting aspects of an individual's microbiome to other metadata. Could the authors try to better connect microbiome structure with isotopic signatures? This might provide better support for the idea that microbiome structure correlates with amino acid metabolism.

Smaller Issues

Mention protein earlier in the introduction. In the 1st paragraph only fiber and toxins are mentioned as challenges associated with plant-based diets, while it's widely known that low protein content of plant material is a common challenge for herbivores.

Line 39+40 – remove this first sentence. The sentence starting “To survive” functions as a good topic sentence.

The paragraph from lines 57-66 seems unnecessary

Line 127 + results: Relabel this as “microbiome inventories” or something similar.

Lines 135 – what version of QIIME? Why 97% OTUs? ASVs are quickly becoming the new standard.

Lines 200-203 – rearrange this statement to something like “We assume that our experiments were in steady state given...”

Line 319 – Why is this a new paragraph? Please insert a better topic sentence.

375 – Significant is spelled incorrectly.

Graphs – do not connect points across amino acids, as the points have no relation to one another.

Decision letter (RSPB-2019-1551.R0)

06-Aug-2019

Dear Dr Newsome:

I am writing to inform you that your manuscript RSPB-2019-1551 entitled "Quantifying the role of the gut microbiome in host essential amino acid metabolism" has, in its current form, been rejected for publication in Proceedings B.

This action has been taken on the advice of referees, who have recommended that substantial revisions are necessary. With this in mind we would be happy to consider a resubmission, provided the comments of the referees are fully addressed. However please note that this is not a provisional acceptance.

The resubmission will be treated as a new manuscript. However, we will approach the same reviewers if they are available and it is deemed appropriate to do so by the Editor. Please note that resubmissions must be submitted within six months of the date of this email. In exceptional

circumstances, extensions may be possible if agreed with the Editorial Office. Manuscripts submitted after this date will be automatically rejected.

Sincerely,
Dr Daniel Costa
mailto: proceedingsb@royalsociety.org

Associate Editor

Board Member: 1

Comments to Author:

Your ms got two very divergent reviews. However, both reviews raised some significant issues. The comments raised by the referees are clear. The comments raised by referee 2 are particularly problematic. It is unclear whether these can be adequately addressed. However, I am willing to give you an opportunity to address them.

Reviewer(s)' Comments to Author:

Referee: 1

Comments to the Author(s)

There are many, relatively minor comments on the manuscript. The two large comments are the following.

- 1) The equation reported for Model 1 is incorrect. It is a linear mixing model, not a concentration dependent model. Report the appropriate equation and use it to calculate the dietary end member.
- 2) Explain the deep assumptions underpinning your approach to modeling the microbial end member. In particular, why isn't dietary cellulose (C3) being considered a potential C substrate for microbes? If it is a substrate, then the C end member used here is wrong.
- 3) I found the entire discussion of catabolism from the gut mucosa pretty much intelligible.

Referee: 2

Comments to the Author(s)

This paper set out to quantify the contribution of gut microbes to the essential amino acid metabolism. This physiological function has been greatly overlooked in the microbiome

literature, and so the techniques used here are promising. However, there were several large issues with this experiment.

Larger Issues

Relevance – the introduction of this paper heavily discusses herbivorous animals, but the experiment was conducted in laboratory mice. These animals are highly inbred, and also vendor-purchased mice have extremely different microbial communities from their wild counterparts, which also influences aspects of their physiology.

[https://www.cell.com/cell/pdf/S0092-8674\(17\)31065-6.pdf](https://www.cell.com/cell/pdf/S0092-8674(17)31065-6.pdf)

Related, the paper doesn't discuss much of what has been demonstrated in regards to microbial provisioning of essential amino acids. These processes have been studied before using labeled substrates, and could be mentioned a bit more in the introduction. Better set up how this paper is novel from what has been done (right now the intro just says that this process is poorly understood). I think the info on fiber degradation could be extensively reduced to allow for more discussion of amino acid- related processes.

Communal housing of mice – it is not widely known that co-housed mice share aspects of their microbiome with each other. This cohousing likely reduced inter-individual variation across your treatment groups, and also results in the fact that these individuals are not completely independent units. <https://academic.oup.com/femsre/article/40/1/117/2467665>

Individual physiology not measured – Much of the details used in the (food intake, digestibility, etc) seem to be taken from the literature. However, I wonder how these variables may have differed across treatment groups, and may have influenced the results and conclusions. For example, the digestive system can change its function to promote optimal digestion/absorption: <https://royalsocietypublishing.org/doi/full/10.1098/rspb.2009.2045> I'd like if the authors could explicitly state what variables in their analysis are being taken from the literature, and how variation here could alter their results.

Microbiome data poorly analyzed and poorly connected to amino acid metabolism – The only microbiome results presented are differences in Firmicutes+Bacteroidetes. With microbiome sequencing there is lots of deeper information that can be gleaned. Also, many studies are now connecting aspects of an individual's microbiome to other metadata. Could the authors try to better connect microbiome structure with isotopic signatures? This might provide better support for the idea that microbiome structure correlates with amino acid metabolism.

Smaller Issues

Mention protein earlier in the introduction. In the 1st paragraph only fiber and toxins are mentioned as challenges associated with plant-based diets, while it's widely known that low protein content of plant material is a common challenge for herbivores.

Line 39+40 – remove this first sentence. The sentence starting “To survive” functions as a good topic sentence.

The paragraph from lines 57-66 seems unnecessary

Line 127 + results: Relabel this as “microbiome inventories” or something similar.

Lines 135 – what version of QIIME? Why 97% OTUs? ASVs are quickly becoming the new standard.

Lines 200-203 – rearrange this statement to something like “We assume that our experiments were in steady state given...”

Line 319 – Why is this a new paragraph? Please insert a better topic sentence.

375 – Significant is spelled incorrectly.

Graphs – do not connect points across amino acids, as the points have no relation to one another.

Author's Response to Decision Letter for (RSPB-2019-1551.R0)

See Appendix A.

RSPB-2019-2447.R0

Review form: Reviewer 1 (Paul Koch)

Recommendation

Major revision is needed (please make suggestions in comments)

Scientific importance: Is the manuscript an original and important contribution to its field?

Excellent

General interest: Is the paper of sufficient general interest?

Excellent

Quality of the paper: Is the overall quality of the paper suitable?

Excellent

Is the length of the paper justified?

Yes

Should the paper be seen by a specialist statistical reviewer?

No

Do you have any concerns about statistical analyses in this paper? If so, please specify them explicitly in your report.

Yes

It is a condition of publication that authors make their supporting data, code and materials available - either as supplementary material or hosted in an external repository. Please rate, if applicable, the supporting data on the following criteria.

Is it accessible?

Yes

Is it clear?

Yes

Is it adequate?

Yes

Do you have any ethical concerns with this paper?

No

Comments to the Author

As I began to re-review this paper, it became clear that the authors did not address any of the embedded comments that I made on the PDF. As they conscientiously addressed my few major comments and both the major and minor comments made by the other reviewer (who I guess provided them in a text document keyed to line numbers), I assume my embedded comments were stripped from the PDF the authors received. There were many minor but other non-trivial issues in my PDF comments. Given that those were not addressed, I can't offer any meaningful comments on the re-review.

I had an overall positive assessment of the paper on the first round. If the paper is published based on the comments of the other reviewer and editors, the authors should look through a readable version of my original PDF (attached) with comments intact and address them as they chose.

Review form: Reviewer 2

Recommendation

Major revision is needed (please make suggestions in comments)

Scientific importance: Is the manuscript an original and important contribution to its field?

Acceptable

General interest: Is the paper of sufficient general interest?

Acceptable

Quality of the paper: Is the overall quality of the paper suitable?

Acceptable

Is the length of the paper justified?

Yes

Should the paper be seen by a specialist statistical reviewer?

No

Do you have any concerns about statistical analyses in this paper? If so, please specify them explicitly in your report.

No

It is a condition of publication that authors make their supporting data, code and materials available - either as supplementary material or hosted in an external repository. Please rate, if applicable, the supporting data on the following criteria.

Is it accessible?

Yes

Is it clear?

Yes

Is it adequate?

Yes

Do you have any ethical concerns with this paper?

No

Comments to the Author

Overall, the authors have considerably improved the paper in terms of relevance and flow. However, some considerable issues still exist:

In a response to Reviewer 1, the authors state that they do not include cellulose as a substrate because these processes happen in the hindgut, and so instead they are focusing on small intestinal processes. Moreover, the authors argue that the hindgut is primarily involved in amino acid metabolism, and poor absorption. However, see van der Wielen et al. 2017 (J Nutr), . Last, these processes ignore the fact that mice are coprophagic and ingest their own feces. So, the point from Reviewer 1 might still stand, or requires some responses to these issues.

Along these lines, if the authors are focusing on small intestinal processes, why were microbial inventories conducted from cecal contents? The gut communities can vary considerably across gut regions (<https://journals.plos.org/plosone/article?id=10.1371/journal.pone.0163720>)

Line 70 – this paragraph needs a better topic sentence

The discussion focuses heavily on which taxa were changed by diet, but not which contribute to microbial aa (there is speculation on this, but I cannot glean data to strongly support). The authors set this up as a goal of their paper, by writing “we currently do not know how widespread this phenomenon is and which taxa of bacteria are involved” in the introduction. As it stands, I’m not sure the current paper addresses this knowledge gap. It is true that the authors attempted this through RDA, but then at lines 214-216 there is only a short statement saying that no OTUs were significant. However, I then wonder if this is a sample size issue? The authors state 10 animals per group, but the microbiome graphs (Fig 2) only show 5 per group. Additionally, could there be genera, family, etc that might come out as significant, rather than analyzing at the OTU level?

In Fig 2B the red points are spread out quite a bit on the line. Do those individuals vary at all in their isotopic signatures or microbial contributions?

Related to above, please be specific in the manuscript where you sample sizes might differ from the 60 original mice. It might be useful to put sample sizes in figure legends?

Decision letter (RSPB-2019-2447.R0)

20-Nov-2019

I am writing to inform you that this version of your manuscript RSPB-2019-2447 entitled "Isotopic and genetic methods reveal the role of the gut microbiome in mammalian host essential amino acid metabolism" has, in its current form, been rejected for publication in Proceedings B.

This action has been taken on the advice of referees, who have recommended that substantial revisions are necessary. We do not normally allow multiple rounds of revision, but I am willing to make an exception and consider a resubmission, provided the comments of the referees are fully addressed. However please note that this is not a provisional acceptance.

The resubmission will be treated as a new manuscript. However, we will approach the same reviewers if they are available and it is deemed appropriate to do so by the Editor. Please note

that resubmissions must be submitted within six months of the date of this email. In exceptional circumstances, extensions may be possible if agreed with the Editorial Office. Manuscripts submitted after this date will be automatically rejected.

Please find below the comments made by the referees, not including confidential reports to the Editor, which I hope you will find useful.

Sincerely,
Dr Daniel Costa
mailto: proceedingsb@royalsociety.org

Reviewer(s)' Comments to Author:

Referee: 2

Comments to the Author(s).

Overall, the authors have considerably improved the paper in terms of relevance and flow. However, some considerable issues still exist:

In a response to Reviewer 1, the authors state that they do not include cellulose as a substrate because these processes happen in the hindgut, and so instead they are focusing on small intestinal processes. Moreover, the authors argue that the hindgut is primarily involved in amino acid metabolism, and poor absorption. However, see van der Wielen et al. 2017 (J Nutr), . Last, these processes ignore the fact that mice are coprophagic and ingest their own feces. So, the point from Reviewer 1 might still stand, or requires some responses to these issues.

Along these lines, if the authors are focusing on small intestinal processes, why were microbial inventories conducted from cecal contents? The gut communities can vary considerably across gut regions (<https://journals.plos.org/plosone/article?id=10.1371/journal.pone.0163720>)

Line 70 - this paragraph needs a better topic sentence

The discussion focuses heavily on which taxa were changed by diet, but not which contribute to microbial aa (there is speculation on this, but I cannot glean data to strongly support). The authors set this up as a goal of their paper, by writing "we currently do not know how widespread this phenomenon is and which taxa of bacteria are involved" in the introduction. As it stands, I'm not sure the current paper addresses this knowledge gap. It is true that the authors attempted this through RDA, but then at lines 214-216 there is only a short statement saying that no OTUs were significant. However, I then wonder if this is a sample size issue? The authors state 10 animals per group, but the microbiome graphs (Fig 2) only show 5 per group. Additionally, could there be genera, family, etc that might come out as significant, rather than analyzing at the OTU level?

In Fig 2B the red points are spread out quite a bit on the line. Do those individuals vary at all in their isotopic signatures or microbial contributions?

Related to above, please be specific in the manuscript where you sample sizes might differ from the 60 original mice. It might be useful to put sample sizes in figure legends?

Referee: 1

Comments to the Author(s).

As I began to re-review this paper, it became clear that the authors did not address any of the embedded comments that I made on the PDF. As they conscientiously addressed my few major comments and both the major and minor comments made by the other reviewer (who I guess provided them in a text document keyed to line numbers), I assume my embedded comments were stripped from the PDF the authors received. There were many minor but other non-trivial issues in my PDF comments. Given that those were not addressed, I can't offer any meaningful comments on the re-review.

I had an overall positive assessment of the paper on the first round. If the paper is published based on the comments of the other reviewer and editors, the authors should look through a readable version of my original PDF (attached) with comments intact and address them as they chose.

Author's Response to Decision Letter for (RSPB-2019-2995.R0)

See Appendix B.

RSPB-2019-2995.R1 (Revision)

Review form: Reviewer 1

Recommendation

Accept with minor revision (please list in comments)

Scientific importance: Is the manuscript an original and important contribution to its field?

Excellent

General interest: Is the paper of sufficient general interest?

Excellent

Quality of the paper: Is the overall quality of the paper suitable?

Excellent

Is the length of the paper justified?

Yes

Should the paper be seen by a specialist statistical reviewer?

No

Do you have any concerns about statistical analyses in this paper? If so, please specify them explicitly in your report.

No

It is a condition of publication that authors make their supporting data, code and materials available - either as supplementary material or hosted in an external repository. Please rate, if applicable, the supporting data on the following criteria.

Is it accessible?

No

Is it clear?

No

Is it adequate?

No

Do you have any ethical concerns with this paper?

No

Comments to the Author

I made comments on both the paper (embedded in the pdf) and on the supplementary material. Most are minor. My only significant concern relates to the mixing model used to generate end-member $\delta^{13}\text{C}$ values for individual amino acids on each diet. The formula in the draft I received was wrong, and the description of that formula is not correct either. The authors need to explain how they calculated the proportion of each amino acid coming from casein vs. cornmeal in each diet and ideally provide a table showing these proportions for each amino acid on each diet. They should be accounting for a) the concentration of the casein and cornmeal in each diet, b) the concentration of protein in casein (100%) and cornmeal (10%), and c) the concentration (by weight, not molar fraction) of each amino acid in casein and cornmeal protein. From those data, they could calculate how many grams of each amino acid were ingested (per day or over the course of the whole experiment) from casein and cornmeal protein on each diet (which can be converted to the proportions needed in model 1. Maybe they did this.

Their description of how the authors calculated amino acid supply is very close to what they need to report for model 1. However, that discussion raises another important question. In model 1, did they account for digestibility of each of the amino acids? This seems like an important factor, but it raises a pretty intractable issue. The digestibility of a particular AA in a protein may, itself, be affected by the microbial community. So while it is great to be aware of differences in digestibility (and I guess fine for Fig. 6), it is probably better to just include differences in ingested amino acids rather than "assimilated" amino acids, since digestibility has likely not been measured on such wildly different diets.

Review form: Reviewer 2

Recommendation

Accept with minor revision (please list in comments)

Scientific importance: Is the manuscript an original and important contribution to its field?

Good

General interest: Is the paper of sufficient general interest?

Good

Quality of the paper: Is the overall quality of the paper suitable?

Good

Is the length of the paper justified?

Yes

Should the paper be seen by a specialist statistical reviewer?

No

Do you have any concerns about statistical analyses in this paper? If so, please specify them explicitly in your report.

No

It is a condition of publication that authors make their supporting data, code and materials available - either as supplementary material or hosted in an external repository. Please rate, if applicable, the supporting data on the following criteria.

Is it accessible?

Yes

Is it clear?

Yes

Is it adequate?

Yes

Do you have any ethical concerns with this paper?

No

Comments to the Author

I commend the authors on improving this manuscript. I think it will make a great contribution to the field.

My only remaining comment is that I would appreciate if the authors were more upfront that this study wasn't designed to identify directly which microbes contribute to host AA metabolism. Their statement in the introduction remains "however, we currently do not know how widespread this phenomenon is and which taxa of bacteria are involved." I might suggest removing this portion of the sentence in the intro. Explicitly recognizing this as a remaining open question in the discussion might be more appropriate.

Decision letter (RSPB-2019-2995.R1)

27-Jan-2020

Dear Dr Newsome

I am pleased to inform you that your manuscript RSPB-2019-2995 entitled "Isotopic and genetic methods reveal the role of the gut microbiome in mammalian host essential amino acid metabolism" has been accepted for publication in Proceedings B.

The referee(s) have recommended publication, but also suggest some minor revisions to your manuscript. Therefore, I invite you to respond to the referee(s)' comments and revise your manuscript. Because the schedule for publication is very tight, it is a condition of publication that you submit the revised version of your manuscript within 7 days. If you do not think you will be able to meet this date please let us know.

- DNA sequences: Genbank accessions F234391-F234402

- Phylogenetic data: TreeBASE accession number S9123
- Final DNA sequence assembly uploaded as online supplemental material
- Climate data and MaxEnt input files: Dryad doi:10.5521/dryad.12311

[http://datadryad.org/submit?journalID=RSPB&manu=\(Document not available\)](http://datadryad.org/submit?journalID=RSPB&manu=(Document%20not%20available)) which will take you to your unique entry in the Dryad repository. If you have already submitted your data to dryad you can make any necessary revisions to your dataset by following the above link. Please see <https://royalsociety.org/journals/ethics-policies/data-sharing-mining/> for more details.

Sincerely,
Dr Daniel Costa
mailto:proceedingsb@royalsociety.org

Reviewer(s)' Comments to Author:

Referee: 2

Comments to the Author(s).

I commend the authors on improving this manuscript. I think it will make a great contribution to the field.

My only remaining comment is that I would appreciate if the authors were more upfront that this study wasn't designed to identify directly which microbes contribute to host AA metabolism. Their statement in the introduction remains "however, we currently do not know how widespread this phenomenon is and which taxa of bacteria are involved." I might suggest removing this portion of the sentence in the intro. Explicitly recognizing this as a remaining open question in the discussion might be more appropriate.

Referee: 1

Comments to the Author(s).

I made comments on both the paper (embedded in the pdf) and on the supplementary material. Most are minor. My only significant concern relates to the mixing model used to generate end-member $\delta^{13}C$ values for individual amino acids on each diet. The formula in the draft I received was wrong, and the description of that formula is not correct either. The authors need to explain how they calculated the proportion of each amino acid coming from casein vs. cornmeal in each diet and ideally provide a table showing these proportions for each amino acid on each diet. They should be accounting for a) the concentration of the casein and cornmeal in each diet, b) the concentration of protein in casein (100%) and cornmeal (10%), and c) the concentration (by weight, not molar fraction) of each amino acid in casein and cornmeal protein. From those data, they could calculate how many grams of each amino acid were ingested (per day or over the

course of the whole experiment) from casein and cornmeal protein on each diet (which can be converted to the proportions needed in model 1. Maybe they did this.

Their description of how the authors calculated amino acid supply is very close to what they need to report for model 1. However, that discussion raises another important question. In model 1, did they account for digestibility of each of the amino acids? This seems like an important factor, but it raises a pretty intractable issue. The digestibility of a particular AA in a protein may, itself, be affected by the microbial community. So while it is great to be aware of differences in digestibility (and I guess fine for Fig. 6), it is probably better to just include differences in ingested amino acids rather than "assimilated" amino acids, since digestibility has likely not been measured on such wildly different diets.

Author's Response to Decision Letter for (RSPB-2019-2995.R1)

See Appendix C.

Decision letter (RSPB-2019-2995.R2)

04-Feb-2020

Dear Dr Newsome

I am pleased to inform you that your manuscript entitled "Isotopic and genetic methods reveal the role of the gut microbiome in mammalian host essential amino acid metabolism" has been accepted for publication in Proceedings B.

Open Access

Paper charges

Sincerely,
Proceedings B
mailto:proceedingsb@royalsociety.org

Appendix A

The University of New Mexico

Seth D. Newsome
Associate Professor, Biology Department
Associate Director, Center for Stable Isotopes
190 Castetter Hall
Albuquerque, NM 87131
newsome@unm.edu
<http://snewsome.org>
(831) 566-3276

22 October 2019

Dear Dr. Daniel Costa:

On behalf of my co-authors, I am pleased to re-submit the manuscript "Isotopic and genetic methods reveal the role of the gut microbiome in mammalian host essential amino acid metabolism" for consideration as an Article in the *Proceedings of the Royal Society B* (RSPB-2019-1551). We appreciate the constructive comments provided by both reviewers and have done our best to address them in this extensive revision.

Attached to this cover letter are point-by-point responses to each comment, and we have uploaded both edited and clean versions of our revised main text and accompanying electronic supporting material file for review; all files are editable Word documents. We hope you find the manuscript materials to be in good order and we look forward to your evaluation of our work.

Sincerely,

Seth D. Newsome, Ph.D.

Referee: 1

There are many, relatively minor comments on the manuscript. The two large comments are the following.

1) The equation reported for Model 1 is incorrect. It is a linear mixing model, not a concentration dependent model. Report the appropriate equation and use it to calculate the dietary end member.

Response: We appreciate the comment and we apologize for the mistake. Indeed, the equation for Model #1 in the electronic supplementary material is not a concentration-dependent mixing model, it's a simple linear model that calculates proportion (p) of each essential amino acid (AA_{ESS}) as the product of the amount of each dietary protein (casein or cornmeal) in each diet treatment and the amino acid concentration of each of these sources. This mistake has now been corrected in both the main text and electronic supporting material.

2) Explain the deep assumptions underpinning your approach to modeling the microbial end member. In particular, why isn't dietary cellulose (C_3) being considered a potential C substrate for microbes? If it is a substrate, then the C end member used here is wrong.

Response: The primary reasons we didn't include dietary cellulose as a major source of substrate for microbial amino acid synthesis is because previous research has shown that (1) microbially assisted cellulose fermentation primarily occurs in the large intestine (White et al. 2014), and (2) amino acids are not significantly absorbed by the colonic mucosa and are instead intensively metabolized by the large intestine microbiome (Tome et al. 2013). This is why our schematic represented by Figure 1 only considers processes occurring in the small intestine. While the primary products of cellulose fermentation include short-chained fatty acids (SCFA) that are assimilated by the colonic mucosa, and eventually could be converted into amino acids by the host organisms once SCFA pass across the intestinal wall, our experimental design did not enable us to trace this pathway because cellulose and the primary source of dietary protein (casein) had similar $\delta^{13}C$ values (Table S1). Thus, our calculations of the microbial contribution of individual AA_{ESS} (Figure 4) provide a minimum estimate as they only consider (1) processes occurring in the small intestine where the large majority of (dietary or microbially synthesized) AAs are assimilated by the host, and (2) the conversion of simple (sucrose) and more complex (cornmeal) dietary carbohydrates into AA_{ESS} by the gut microbiome.

3) I found the entire discussion of catabolism from the gut mucosa pretty much intelligible.

Response: We assume that the reviewer intended to say "unintelligible" for this comment. It's not clear which section(s) of the manuscript the reviewer is referring to here but in response to questions raised by Reviewer #2 about the assumptions that we included in our calculations for AA supply and demand, we now provide a more comprehensive description of gut mucosal metabolism in the subsection entitled "Relative AA_{ESS} Supply and Demand" at the end of the Methods section (Lines 199–216) to clarify the assumptions that we used to calculate the data reported in Figure 6. If the reviewer is referring to the paragraph near the end of the Discussion that mentioned gut mucosal catabolism but mainly focused on the potential role of eukaryotic gut microbes, we have removed this paragraph from the revised manuscript as it did not significantly add to the background of the project given our results.

Referee: 2

This paper set out to quantify the contribution of gut microbes to the essential amino acid metabolism. This physiological function has been greatly overlooked in the microbiome literature, and so the techniques used here are promising. However, there were several large issues with this experiment.

Response: We agree with the reviewer that the physiological function of the gut microbiome in general, and more specifically its contribution to the protein metabolism of the host has been largely overlooked in the literature. And we acknowledge there are aspects of our experiment that do not make it a perfect analog for free-ranging small (or large) herbivorous or omnivorous mammals. However, our results and the approach we used in this initial experiment is an important first step to examining these processes in wild mammal populations, and we chose to study a lab-reared mammalian omnivore that is commonly used in the biomedical field because we figured it would provide an initial minimum estimate of the importance of this pathway (microbial AA_{ESS} provisioning) in host protein metabolism. Interestingly, we tackled many of the larger issues identified by this reviewer (e.g., communal housing of mice, species selection) when designing a series of future experiments that were recently funded by NSF-DIOS. But there is little we can do at this time to address the choice of model organism. As suggested, we have toned down the potential implications for wild species, and a re-analysis of our data show that some of the issues raised by this reviewer (e.g., communal housing) do not homogenize the microbial community composition, or the apparent degree of microbial provisioning of AA_{ESS} among individual hosts within a diet treatment. We have added short sections to both the main text and the Electronic Supporting Material (ESM) that addresses these issues with additional analyses. Specifically, we now highlight the large degree of variation in both microbiome composition and contribution to host AA metabolism within diet treatments (see below). In regards to the issues related to individual physiology, below we provide further explanation of our assumptions used to calculate AA_{ESS} supply and demand and have extensively modified the main text of the manuscript to clarify our approach and assumptions. We also note that the supply and demand calculations were used as an interpretive tool as a first order estimate of a potential factor that could explain observed variation in the degree of microbial contribution to the AA_{ESS} pool used by the host to build tissues. Lastly, in response to the final larger issue raised by this reviewer, we have re-analyzed our genetic data and now present more information about patterns in the gut microbiome at various levels of taxonomic organization within and among diet treatments.

Larger Issues

Relevance – the introduction of this paper heavily discusses herbivorous animals, but the experiment was conducted in laboratory mice. These animals are highly inbred, and also vendor-purchased mice have extremely different microbial communities from their wild counterparts, which also influences aspects of their physiology. [https://www.cell.com/cell/pdf/S0092-8674\(17\)31065-6.pdf](https://www.cell.com/cell/pdf/S0092-8674(17)31065-6.pdf)

*Response: Indeed, a limitation of our experiment is that it used a veritable Frankenstein model species, which likely have different gut microbiome communities than the wild mice we capture in the northern Chihuahuan Desert of New Mexico. When we began this line of research, we did not have the ability to raise wild species of omnivorous mice (e.g., *Peromyscus* spp.) in the laboratory and thus chose to begin with commercially available house mice (*Mus musculus*), a model organism in the biomedical sciences that we could easily house and raise in our animal research facility. Despite this limitation, we contend our data for house mice do provide novel insights into the provisioning of essential amino acids from the gut microbiome to a mammalian host, which to our knowledge has not been quantified for the entire suite of major essential amino acids in a single experiment/organism; we term this set ‘major’ because they represent ~95% of the essential amino acids in house mouse muscle tissue (Wolf et al. 2015). The only other studies that has attempted this with a large suite of AA_{ESS} have focused on terrestrial insects with unique dietary (xylophagous) niches (Ayayee et al. 2015a, 2015b). As such, we contend that our data provide a novel baseline for future experiments on other herbivorous, omnivorous, and even carnivorous mammals whose gut microbiomes may play important roles in converting dietary macromolecules (carbohydrates or lipids) into AA_{ESS} that are ultimately “shared” with the host and used to maintain protein homeostasis.*

*To address this comment, we have toned down our mention of herbivores and give equal attention to omnivores, but still acknowledge that this process is likely important for both functional groups of consumers. We have also inserted a statement in the Experimental Design sub-section of the Methods (Lines 135–137) to acknowledge that house mice are not ideal analogs for wild omnivorous mice such as *Peromyscus*, but these two species do share the same fundamental biochemical pathways important for amino acid metabolism.*

Related, the paper doesn't discuss much of what has been demonstrated in regards to microbial provisioning of essential amino acids. These processes have been studied before using labeled substrates, and could be mentioned a bit more in the introduction. Better set up how this paper is novel from what has been done (right now the intro just says that this process is poorly understood). I think the info on fiber degradation could be extensively reduced to allow for more discussion of amino acid- related processes.

Response: We appreciate the suggestion and agree that a better description of isotope labeling experiments is a good addition to the Introduction to contextualize our experiment, and highlight how our multi-amino acid approach based on isotopic analysis at natural abundances is unique in comparison to this body of literature. We have revised the paragraph in the Introduction (Lines 85-99) to include this information. Our revision builds on the series of labelling-based studies papers that were cited in our initial submission (Torrallardona et al. 1996a, 1996b, Metges et al. 1999, Metges 2000), and does a better job of describing the role that the gut microbiome plays in the digestibility of dietary protein; additional details can be found in our response to the comment below regarding individual digestive physiology.

Communal housing of mice – it is not widely known that co-housed mice share aspects of their microbiome with each other. This cohousing likely reduced inter-individual variation across your treatment groups, and also results in the fact that these individuals are not completely independent units. <https://academic.oup.com/femsre/article/40/1/117/2467665>

Response: We agree with the reviewer that communal housing of the mice would buffer differences among the mice within a cage or could potentially result in a cage effect. However, we observed significant treatment effects by diet in both microbiome composition (Figure 2D) and modeled estimates of the microbiome contribution to protein metabolism (Figure 4) despite this caveat. In regards to observed variation in AA $\delta^{13}\text{C}$ values, standard deviations within treatments varied from as little as 0.7‰ to ~3.5‰, which corresponds to variation of up to 30% in the estimated microbial contribution among individual mice in a single diet treatment given the (15–20‰) spread in $\delta^{13}\text{C}$ values of dietary protein versus microbially synthesized AAs available to mice (Figure 3). In regards to microbiome composition, mice co-housed in this experiment did not develop homogenous microbiota as reported in the article (and references within) suggested above by the reviewer. The family level community compositional differences among the mice are noteworthy and can be seen in panel D of Figure 2 (diversity differences are shown in Figure S2) of the original manuscript. We have now performed a Monte-Carlo significance test that indicates that the OTU level differences of the mouse gut microbiota between individual mice within each diet are largely significant (p -values < 0.01 for at least 80% of the pairwise comparisons of the individuals within a diet, except in Diet 1, where only 50% were significant). Thus, because we observed such a strong diet effect (e.g., ratio of baseline error to observed error in the random forests analysis was 15 when a minimum ratio of 2 is expected for factors that can be accurately predicted) and found that pairwise comparisons of sample distances revealed greater variation of the gut microbiota among than within diet treatments (among diet average = 0.67 +/- 0.14, within diet average = 0.54 +/- 0.14; Kruskal-Wallis $P < 0.001$), we do not believe that a cage effect has contributed to our findings. In addition, although co-housing the mice may compromise statistical independence, we used non-parametric statistics to test for treatment effects, which do not assume independence. We thank the reviewer for raising this important issue and we have now added text to the ESM (Gut Microbiome Community Composition, Page 1) recognizing this caveat.

Individual physiology not measured – Much of the details used in the (food intake, digestibility, etc) seem to be taken from the literature. However, I wonder how these variables may have differed across treatment groups, and may have influenced the results and conclusions. For example, the digestive system can change its function to promote optimal digestion/absorption: <https://royalsocietypublishing.org/doi/full/10.1098/rspb.2009.2045> I'd like if the authors could explicitly state what variables in their analysis are being taken from the literature, and how variation here could alter their results.

Response: The reviewer is correct, our analysis reported in Figure 5 leans on published estimates of protein digestibility for mammals (pigs, rats, and mice) to estimate the relative supply of amino acids in each diet versus demand. We now explicitly state this in the Methods (Lines 199–202) and do a better job of describing the two-component approach we used to make the supply (digestibility + catabolism) calculations in the subsection “Relative AA_{Ess} Supply versus Demand” in the Methods (Lines 199–216) section of the main text and in the subsection of the same name in the ESM. Our approach is also summarized in the following paragraphs of this response:

The primary objective of this exercise was to help interpret the AA-specific patterns of microbiome contribution shown in Figure 4. Our digestibility calculations show that the two diet treatments containing the highest protein contents, 21% (Diet #2) and 40% (Diet #1), provided an adequate amount of amino acids, while estimates for the other two diets containing less protein were similar to (Diet #3) or slightly lower than (Diet #4) demand for all six essential amino acids we measured. Thus, we might expect that the gut microbiome contribution to host protein metabolism in diets containing relatively low amounts of protein would be higher than in the diets containing more protein, which is exactly what we observed for several amino acids (e.g., Val, Leu, Ile, Thr). Another perhaps more striking conclusion of these calculations is the gut microbiome’s contribution is still large (10–30%) for many essential AAs when mice are fed diets (Diet #1 and #2) containing more than adequate amounts of amino acids to meet demand.

*In our revision of the Methods subsection titled “Relative AA_{Ess} Supply versus Demand” we do a better job of describing the primary controls on digestibility, which we distill into two primary components (Metges 2000, Neis et al. 2015). The first factor that influences AA supply is associated with uptake efficiency of different sources of dietary protein (casein and cornmeal), and we used AA-specific estimates for *Mus musculus* reported in Keith and Bell (1988) to account for this component; we include these data in Table S7 for the essential amino acids that are the focus of our study. In Table 5 of Keith and Bell (1988), they report AA-specific estimates of true digestibility for diets containing 15% and 30% protein for a variety of protein sources (including casein and zein). AA-specific digestibility estimates are generally within ~5% for these two protein concentrations. While our diets have a slightly broader range of protein concentrations (10 to 40%), we believe that the estimates provided in Keith and Bell (1988) are suitable for these calculations.*

The second factor that influences AA supply is catabolism of amino acids by the gut microbiome, which ultimately decreases the amino acid availability (size) of the pool of AAs available to the host for digestion. We used estimates in the review by Wu (1998) to account for this component, which is largely based on pigs but also includes data for other domesticated mammals and humans. Not that estimates of mucosal catabolism for specific AAs does not wildly vary among mammalian host species. While not specifically acknowledged by Wu (1988) as this paper preceded the recognition of the important role the gut microbiome plays in macromolecular nutrition, microbes (largely bacteria) are likely responsible for the majority of mucosal catabolism in mammalian guts. When synthesizing this literature, we were surprised to find that mucosal catabolism could reduce the intestinal amino acid pool available for digestion by as much as 30% (Ile) to 70% (Phe), and overall has a much larger impact on AA availability for host assimilation than true digestibility of specific AAs as discussed above. But after consideration of this process, supply in most diets still exceeded demand for mice that were actually growing faster than those in our experiment (John and Bell 1976).

Microbiome data poorly analyzed and poorly connected to amino acid metabolism – The only microbiome results presented are differences in Firmicutes+Bacteroidetes. With microbiome sequencing there is lots of deeper information that can be gleaned. Also, many studies are now connecting aspects of an individual's microbiome to other metadata. Could the authors try to better connect microbiome structure with isotopic signatures? This might provide better support for the idea that microbiome structure correlates with amino acid metabolism.

We have revised the manuscript to include additional information about the microbiome and to highlight more clearly the dense, but concise results we presented in the original manuscript. It should be noted though that while we mostly describe our results at the phylum level, our analysis was done at the "species" level. We focus on the Firmicutes vs. Bacteroidetes in this manuscript because these are two major gut phyla that have previously been investigated with respect to carbohydrate metabolism. In addition, with over 8000 OTUs (after rarefaction, or +30,000 without), it is outside the scope of the objectives of this manuscript to discuss much more than family level differences among the different diets we tested. We have now highlighted two important mucin related species that we found in our analysis and performed additional analyses that go beyond the Firmicutes:Bacteroidetes ratio to directly link the diversity data to the stable isotope results. Finally, we are hesitant to discuss more than the associations we observed among these OTUs, the diets, and the stable isotope data in the absence of more meaningful genomic data because 16S rRNA gene phylogeny is often a poor predictor of function.

Smaller Issues

Mention protein earlier in the introduction. In the 1st paragraph only fiber and toxins are mentioned as challenges associated with plant-based diets, while it's widely known that low protein content of plant material is a common challenge for herbivores.

Response: We appreciate the suggestion and now have added "low-protein content" in the first paragraph of the Introduction as a primary reason why many plant tissues are considered to be low quality for consumers.

Line 39+40 – remove this first sentence. The sentence starting "To survive" functions as a good topic sentence.

Response: We appreciate the suggestion and have revised the manuscript accordingly.

The paragraph from lines 57-66 seems unnecessary

Response: We appreciate the suggestion, but respectfully disagree with the reviewer. We think it's important to briefly describe the recognized roles of the two dominant bacteria phyla in mammalian guts. We also believe that it's important to highlight the fact that the majority of research on their functional roles has focused on carbohydrate metabolism, because most of that research is driven by biomedical studies focused on humans. This helps set up the gap in our understanding of their role in protein metabolism, which is the focus of the next paragraph and our study in general.

Line 127 + results: Relabel this as "microbiome inventories" or something similar.

Response: We have relabeled this subsection "Microbiome Community Composition"

Lines 135 – what version of QIIME? Why 97% OTUs? ASVs are quickly becoming the new standard.

QIIME version 1.9.1 was used for our analyses and we have noted this in the revised manuscript. We used 97% DNA:DNA similarity as our OTU criterion because in the absence of any meaningful ecological/evolutionary data to point us to the ASV criterion in this system, we believe it is less arbitrary than actual sequence variants. The 97% OTU criterion is at least based on the gold standard definition of a prokaryotic species (70% DNA:DNA hybridization) and is a more conservative estimate of diversity. It has been said that a single OTU can represent multiple "species", but it should be noted that a single

organism can be represented by multiple ASVs. Given that the Firmicutes are infamous for having multiple rRNA copies per species, we believe that OTUs is a more conservative estimate of diversity for this dataset. Also, although some research groups are decided on using the ASV criterion, there is still significant hesitation among the community to switch. Although some research groups are decided on using the ASV criterion, there is still significant hesitation among the community to switch. For comparison, we did analyze our data using the ASV criterion and found that regardless of whether we rarefied our dataset, the samples clustered by diet more strongly when we used the 97% OTU criterion (see below). We have revised the manuscript to report the number ASVs in the dataset. The figure below has now been included in the Electronic Supplemental Material (Figure S2).

Lines 200-203 – rearrange this statement to something like “We assume that our experiments were in steady state given...”

Response: We appreciate the suggestion and have revised the manuscript accordingly.

Line 319 – Why is this a new paragraph? Please insert a better topic sentence.

Response: We appreciate the suggestion and have extensively revised the Discussion to better organize this section by distinct groups of AA_{ESS} (e.g., branch-chained amino acids) based on their metabolic precursors.

375 – Significant is spelled incorrectly.

Response: We appreciate the correction and have revised the manuscript accordingly.

Graphs – do not connect points across amino acids, as the points have no relation to one another.

Response: We included these lines for graphical clarity to help readers distinguish differences within and among treatments, which was stated in the caption for Figures 3 and 4 of the original submission. We have removed these lines in the latter figure (now Figure 5) in accordance with the reviewer's suggestion, but we have kept them in the former figure (now Figure 4) because they nicely show how the mouse muscle essential amino acid $d^{13}C$ values lie between the two end-member sources: dietary protein and microbially derived amino acids synthesized from dietary carbohydrates. As we had done in the original submission, the short sentence in the caption for Figure 4 that states these lines are included for graphical clarity and have no statistical basis remains.

Appendix B

The University of New Mexico

Seth D. Newsome
Associate Professor, Biology Department
Associate Director, Center for Stable Isotopes
190 Castetter Hall
Albuquerque, NM 87131
newsome@unm.edu
<http://snewsome.org>
(831) 566-3276

23 December 2019

Dear Dr. Daniel Costa:

On behalf of my co-authors, I am pleased to re-submit the manuscript "Isotopic and genetic methods reveal the role of the gut microbiome in mammalian host essential amino acid metabolism" for consideration as an Article in the *Proceedings of the Royal Society B* (RSPB-2019-1551). We appreciate the constructive comments provided by both reviewers and have done our best to address them in this extensive revision.

Attached to this cover letter are point-by-point responses to each comment, and we have uploaded both edited and clean versions of our revised main text and accompanying electronic supporting material file for review; all files are editable Word documents. We hope you find the manuscript materials to be in good order and we look forward to your evaluation of our work.

Sincerely,

Seth D. Newsome, Ph.D.

Referee: 2

Overall, the authors have considerably improved the paper in terms of relevance and flow. However, some considerable issues still exist:

In a response to Reviewer 1, the authors state that they do not include cellulose as a substrate because these processes happen in the hindgut, and so instead they are focusing on small intestinal processes. Moreover, the authors argue that the hindgut is primarily involved in amino acid metabolism, and poor absorption. However, see van der Wielen et al. 2017 (J Nutr). Last, these processes ignore the fact that mice are coprophagic and ingest their own feces. So, the point from Reviewer 1 might still stand, or requires some responses to these issues.

Along these lines, if the authors are focusing on small intestinal processes, why were microbial inventories conducted from cecal contents? The gut communities can vary considerably across gut regions (<https://journals.plos.org/plosone/article?id=10.1371/journal.pone.0163720>)

Response: First, please see our response (and associated references) to Reviewer #1's comment below regarding where amino acids are primarily assimilated in the gastrointestinal tract. The reviewer is correct, we did not include cellulose as a potential substrate for microbially synthesized AA_{ESS} in the hindgut, primarily because cellulose and protein had similar $\delta^{13}\text{C}$ values (Table S1) and thus we couldn't reliably trace this pathway. As such, it could be argued that our approach provides a minimum estimate of the microbial contribution of AA_{ESS} to host amino acid metabolism if microbes in the hindgut are using ^{13}C -depleted cellulose as a substrate to synthesize AA_{ESS}. But given that this region of the gut does not (1) host an efficient array of amino acid and peptide transporters, and (2) absorb a significant amount of amino acids in mature animals, we feel that our focus on pathways associated with the small intestine provides a robust estimate of the microbial contribution of individual AA_{ESS} to host tissue synthesis.

In regards to our focus on cecal contents, we targeted this region of the gut as it is directly adjacent (downstream) of the small intestine and (as shown in the study cited by the reviewer) typically harbors some of the highest levels of bacterial diversity in the gastrointestinal tract of house mice (Suzuki and Nachman 2016).

Line 70 – this paragraph needs a better topic sentence

Response: We appreciate the suggestion and have added the sentence: "The potential sources of amino acids and the pathways by which they can be assimilated by the host are numerous (Fig. 1)." to the beginning of this paragraph.

The discussion focuses heavily on which taxa were changed by diet, but not which contribute to microbial aa (there is speculation on this, but I cannot glean data to strongly support). The authors set this up as a goal of their paper, by writing "we currently do not know how widespread this phenomenon is and which taxa of bacteria are involved" in the introduction. As it stands, I'm not sure the current paper addresses this knowledge gap. It is true that the authors attempted this through RDA, but then at lines 214-216 there is only a short statement saying that no OTUs were significant. However, I then wonder if this is a sample size issue? The authors state 10 animals per group, but the microbiome graphs (Fig 2) only show 5 per group. Additionally, could there be genera, family, etc that might come out as significant, rather than analyzing at the OTU level?

Response: In the experiment described here, we are unable to identify directly which microbes contribute to host AA metabolism. We believe that given the vast diversity of microbes detected, even after only considering actual sequence variants (which reduces community complexity substantially), that it is not possible to make a direct connection between specific OTUs and the AA_{ESS} patterns we observed under the different diets. We do not believe this is a sample size issue (n=5 per treatment for the 16S rRNA gene analysis) because as we show in the manuscript, we were able to detect significant community variation among individuals in the experiment Fig. 2D), enabling us to detect specific and robust experimental effects that are in line with our hypothesis of the important role of the gut microbiome under

low protein conditions. Specifically, the results from our RDA analysis were not completely insignificant. Rather, we report that at the OTU level (and genus/family too, though not reported), we do see a significant correlation of the microbial communities and muscle AA_{ESS} $\delta^{13}\text{C}$ values in the two low protein diets, supporting our hypothesis that microbial contribution to host muscle synthesis increases when the supply of exogenous protein is low. Directly linking specific microbial populations to host muscle AA_{ESS} $\delta^{13}\text{C}$ values remains a focus of our current research and we are employing more precise methods of comparing microbial communities to host metabolism.

In Fig 2B the red points are spread out quite a bit on the line. Do those individuals vary at all in their isotopic signatures or microbial contributions?

Response: The reviewer makes a keen observation as there are several individuals in the two low-protein diet treatments (Diet #3/4) that have a more even mixture of Firmicutes and Bacteroidetes in their ceca (Fig. 2B). We can group the individuals fed either Diet #3 or Diet #4 that had a more even mixture of Firmicutes and Bacteroidetes in their ceca (n=4) and compare them to individuals from these two treatments whose ceca were dominated by Firmicutes (n=6). Interestingly, the mice in these two treatments whose ceca were dominated by Firmicutes had slightly, but not significantly, lower $\delta^{13}\text{C}$ for all AA_{ESS} we measured, suggesting a greater degree of direct routing of dietary proteins, than the individuals in these two treatments that had a more even mixture of Firmicutes and Bacteroidetes. However, the $\delta^{13}\text{C}$ values of AA_{ESS} of mice in the low-protein diet treatments whose ceca were dominated by Firmicutes or had a more even mixture of Firmicutes and Bacteroidetes were significantly higher than those in mice fed high-protein diets (Diets #1/2). Overall, these patterns suggest that a degree of individual-level variation in the microbiome and physiology of co-housed mice fed the same diet, which we highlighted in our responses to the first round of reviews.

Related to above, please be specific in the manuscript where you sample sizes might differ from the 60 original mice. It might be useful to put sample sizes in figure legends?

Response: We appreciate the suggestion and have revised the manuscript accordingly. Sample sizes are now presented in the caption for Figures 2, 4, and 5; we also include this information in the caption for Table S4.

Referee: 1

As I began to re-review this paper, it became clear that the authors did not address any of the embedded comments that I made on the PDF. As they conscientiously addressed my few major comments and both the major and minor comments made by the other reviewer (who I guess provided them in a text document keyed to line numbers), I assume my embedded comments were stripped from the PDF the authors received. There were many minor but other non-trivial issues in my PDF comments. Given that those were not addressed, I can't offer any meaningful comments on the re-review. I had an overall positive assessment of the paper on the first round. If the paper is published based on the comments of the other reviewer and editors, the authors should look through a readable version of my original PDF (attached) with comments intact and address them as they chose.

Response: We apologize for not including revisions and responses to Reviewer #1's comments that were embedded in the marked-up PDF referred to here, but this file was not included in the email from the editorial office, nor was it available through Manuscript Central in the Author Centre. After receiving the first round of reviews, we had to request it from the Editorial Office. Our resubmitted manuscript includes revisions to these original comments, and we appreciate the reviewer's thorough review. There are two major concerns outlined in this reviewer's marked-up PDF that we would like to address here since we didn't address these important comments in the first round of reviews:

First, we now include a paragraph in the ESM that describes the assumptions of our mixing model, specifically the AA-specific fractionation factors that were used to estimate the $\delta^{13}\text{C}$ values of AA_{ESS} synthesized by gut bacteria using dietary carbohydrates as the primary carbon source. We acknowledge that this a caveat of our study, however, we do point out that the fractionation factors for Actinobacteria

reported in Larsen et al. 2013 are very similar to those reported by Abelson and Hoering (1961) for Proteobacteria (*Escherichia coli*). Given the difficulty in culturing (anaerobic) gut bacteria and the limited amount of data available in the literature for AA-specific fractionation factors for a variety of bacteria (even at the phyla level), we chose to use the data from Larsen et al. 2013 to estimate the $\delta^{13}\text{C}$ values of AA_{ESS} synthesized by gut bacteria using dietary carbohydrates as the primary carbon source.

Second, while it's possible that bacteria in the hindgut (large intestine) produce AA_{ESS} that are assimilated by the host in this region of the gut, previous work shows that the small intestine is the major site of microbially-produced amino acid absorption (Torrallardona et al. 1996; Metges et al. 1999; Stoll et al. 1998, Millward et al. 2000); depicted by the R_M pathway in Figure 1. In contrast, the primary products of hindgut fermentation for non-ruminants like mice are short-chained fatty acids (Flint et al. 2012, Koropatkin et al. 2012, White et al. 2014), which could be assimilated by the host via the large intestine, but this substrate cannot be transformed by the host into AA_{ESS}. Furthermore, the large intestine does not host an efficient array of amino acid and peptide transporters, and it is generally accepted that the colon does not absorb a significant amount of amino acids in mature animals (Darragh et al. 1994). We now include a brief description in the main text (Lines 55–56) of a potential pathway for short-chained fatty acids synthesized by bacteria in the hindgut (large intestine) to be used by the host (mouse) for energy or to synthesize other metabolites (e.g., non-essential amino acids).

References

- Abelson PH, Hoering TC (1961) Carbon isotope fractionation in formation of amino acids by photosynthetic organisms. *Proceedings of the National Academies of Sciences USA* 47(5):623–632.
- Darragh AJ, Cranwell PD, Moughan PJ (1994) Absorption of lysine and methionine from the proximal colon of the piglet. *British Journal of Nutrition* 71: 739–752.
- Flint HJ, Scott KP, Duncan SH, Louis P, Forano E (2012) Microbial degradation of complex carbohydrates in the gut. *Gut Microbes* 3: 289–306.
- Koropatkin NM, Cameron EA, Martens EC (2012) How glycan metabolism shapes the human gut microbiota. *Nature Reviews Microbiology* 10: 323–335.
- Larsen T, Ventura M, Andersen N, O'Brien DM, Piatkowski U, McCarthy, MD (2013) Tracing carbon sources through aquatic and terrestrial food webs using amino acid stable isotope fingerprinting. *PLoS One* 8(9): e73441.
- Metges CC, Petzke KJ, El-Khoury AE, Henneman L, Grant I, Bedri S, Regan MM, Fuller MF, Young VR (1999) Incorporation of urea and ammonia nitrogen into ileal and fecal microbial proteins and plasma free amino acids in normal men and ileostomates. *American Journal of Clinical Nutrition* 70: 1046–58.
- Millward DJ, Forrester T, Ah-Sing E, Yeboah N, Gibson N, Badaloo A, Boyne M, Reade M, Persaud C, Jackson A (2000) The transfer of ^{15}N from urea to lysine in the human infant. *British Journal of Nutrition* 83: 505–512.
- Stoll B, Henry J, Reeds PJ, Yu H, Jahoor F, Burrin DG (1998) Catabolism dominates the first-pass intestinal metabolism of dietary essential amino acids in milk protein-fed piglets. *The Journal of Nutrition* 128(3): 606–614.
- Torrallardona D, Harris CI, Coates ME, Fuller MF (1996) Microbial amino acid synthesis and utilization in rats: incorporation of ^{15}N from $^{15}\text{NH}_4\text{Cl}$ into lysine in the tissues of germ-free and conventional rats. *British Journal of Nutrition* 76: 689–700.
- White BA, Lamed R, Bayer EA, Flint HJ (2014) Biomass utilization by gut microbiomes. *Annual Review of Microbiology* 68:279–296.

Appendix C

The University of New Mexico

Seth D. Newsome
Associate Professor, Biology Department
Associate Director, Center for Stable Isotopes
190 Castetter Hall
Albuquerque, NM 87131
newsome@unm.edu
<http://snewsome.org>
(831) 566-3276

3 February 2020

Dear Dr. Daniel Costa:

On behalf of my co-authors, I am pleased to re-submit the manuscript "Isotopic and genetic methods reveal the role of the gut microbiome in mammalian host essential amino acid metabolism" for consideration as an Article in the *Proceedings of the Royal Society B* (RSPB-2019-1551). We appreciate the constructive comments provided by both reviewers and have done our best to address them in this extensive revision.

Attached to this cover letter are point-by-point responses to each comment, and we have uploaded both edited and clean versions of our revised main text and accompanying electronic supporting material file for review; all files are editable Word documents. We hope you find the manuscript materials to be in good order and we look forward to your evaluation of our work.

Sincerely,

Seth D. Newsome, Ph.D.

Referee #2

Comments to the Author(s).

I commend the authors on improving this manuscript. I think it will make a great contribution to the field.

My only remaining comment is that I would appreciate if the authors were more upfront that this study wasn't designed to identify directly which microbes contribute to host AA metabolism. Their statement in the introduction remains "however, we currently do not know how widespread this phenomenon is and which taxa of bacteria are involved." I might suggest removing this portion of the sentence in the intro. Explicitly recognizing this as a remaining open question in the discussion might be more appropriate.

Response: We have revised the manuscript accordingly and removed this portion of the sentence in the Introduction.

Referee #1

Comments to the Author(s).

I made comments on both the paper (embedded in the pdf) and on the supplementary material. Most are minor. My only significant concern relates to the mixing model used to generate end-member d13C values for individual amino acids on each diet. The formula in the draft I received was wrong, and the description of that formula is not correct either. The authors need to explain how they calculated the proportion of each amino acid coming from casein vs. cornmeal in each diet and ideally provide a table showing these proportions for each amino acid on each diet. They should be accounting for a) the concentration of the casein and cornmeal in each diet, b) the concentration of protein in casein (100%) and cornmeal (10%), and c) the concentration (by weight, not molar fraction) of each amino acid in casein and cornmeal protein. From those data, they could calculate how many grams of each amino acid were ingested (per day or over the course of the whole experiment) from casein and cornmeal protein on each diet (which can be converted to the proportions needed in model 1. Maybe they did this.

Response: We appreciate the additional detailed edits and have revised Figure 1 as suggested. We have also revised the formula accordingly and now cite Figure S3 in the ESM in the section titled Gut Microbiome Community Composition. Note that gut bacterial diversity and community composition is also discussed in the main text. We did exactly what the reviewer describes above but only described it in text in the main manuscript and ESM, and the concentration (by weight) of each amino acid in casein and cornmeal protein was provided in Table S2. We appreciate the suggestion of adding a table (new Table S3) to the ESM that provides a step-by-step description of how the proportion of each amino acid sourced from casein versus cornmeal was estimated for each diet treatment. Accompanying text has also been added to the ESM section titled Mixing Models.

Their description of how the authors calculated amino acid supply is very close to what they need to report for model 1. However, that discussion raises another important question. In model 1, did they account for digestibility of each of the amino acids? This seems like an important factor, but it raises a pretty intractable issue. The digestibility of a particular AA in a protein may, itself, be affected by the microbial community. So while it is great to be aware of differences in digestibility (and I guess fine for Fig. 6), it is probably better to just include differences in ingested amino acids rather than "assimilated" amino acids, since digestibility has likely not been measured on such wildly different diets.

Response: No, we did not account for digestibility of each amino acid in Model #1 because as the reviewer suggests, it raises more issues than it solves given our lack of understanding of digestibility for the specific diets used in our experiment. This was a motivating factor for performing the supply and demand calculations shown in Figure 6 where we used digestibility estimates based on diets containing 15% and 30% casein (Keith and Bell 1988), which is within the range used in our experiments. Note that AA-specific estimates of digestibility reported in Keith and Bell (1988) did not vary much between the diets containing 15% and 30% casein.